# CLIPLoss and Norm-Based Data Selection Methods for Multimodal Contrastive Learning

**Yiping Wang**[*]
University of Washington

**Yifang Chen**[*]
University of Washington

**Wendan Yan**
University of Washington

**Alex Fang**
University of Washington

**Wenjing Zhou**
University of Michigan

**Kevin Jamieson**
University of Washington

**Simon Shaolei Du**
University of Washington

## Abstract

Data selection has emerged as a core issue for large-scale visual-language model pretraining (e.g., CLIP), particularly with noisy web-curated datasets. Three main data selection approaches are: (1) leveraging external non-CLIP models to aid data selection, (2) training new CLIP-style embedding models that are more effective at selecting high-quality data than the original OpenAI CLIP model, and (3) designing better metrics or strategies universally applicable to any CLIP embedding without requiring specific model properties (e.g., CLIPScore is one popular metric). While the first two approaches have been extensively studied, the third remains under-explored. In this paper, we advance the third approach by proposing two new methods. Firstly, instead of classical CLIP scores that only consider the alignment between two modalities from a single sample, we introduce **negCLIPLoss**, a method inspired by CLIP training loss that adds the alignment between one sample and its contrastive pairs as an extra normalization term to CLIPScore for better quality measurement. Secondly, when downstream tasks are known, we propose a new norm-based metric, **NormSim**, to measure the similarity between pretraining data and target data. We test our methods on the data selection benchmark, DataComp [1]. Compared to the best baseline using only OpenAI's CLIP-L/14, our methods achieve a 5.3% improvement on ImageNet-1k and a 2.8% improvement on 38 downstream evaluation tasks. Moreover, both **negCLIPLoss** and **NormSim** are compatible with existing techniques. By combining our methods with the current best methods DFN [2] and HYPE [3], we can boost average performance on downstream tasks by 0.9%, achieving a new state-of-the-art on the DataComp-medium benchmark[2].

## 1 Introduction

Curating large-scale visual-language datasets from web-sourced data has become common for pretraining multi-modal models. However, the quality of these web-curated data pairs remains a critical bottleneck. Research has shown that the choice of dataset significantly impacts model performance, irrespective of the models and training techniques employed [4–11], and this motivates

---

[*]Equal contribution. Correspondence to `ypwang61@cs.washington.edu`. Codes are available at `https://github.com/ypwang61/negCLIPLoss_NormSim`.

[2]DataComp benchmark: `https://www.datacomp.ai/dcclip/leaderboard.html`.

38th Conference on Neural Information Processing Systems (NeurIPS 2024).

the development of various data selection strategies. This paper focuses on optimizing subset selection from a fixed data pool to train a CLIP model [4] that achieves superior performance on zero-shot downstream tasks.

Classical methods *rely solely on OpenAI's (OAI) pretrained CLIP model* (i.e., a teacher model) and focus on better utilizing the embeddings. The most commonly used one is calculating CLIPScore, which measures the cosine similarity between the visual and language embeddings of the CLIP model for the same sample, to eliminate low-quality data with mismatches between text and image. Other works also leverage heuristic distribution alignment techniques to select samples relevant to downstream tasks, such as image-based filtering [1]. These approaches are generally viewed as providing only limited enhancements. However, we argue that the potential of those embeddings has been heavily under-explored. This work seeks a universal method to better employ any given embeddings, not only from OAI CLIP, but also from other CLIP-style models.

On the other hand, recent leading data filtering methods, instead of focusing on improving embedding utilization stategy itself, mainly follow the other two directions, both employing external resources. They either (1) use *external non-CLIP models* that aid in data selection, (2) or use *external high-quality multi-modal data* to train a *better CLIP-style embedding model* than the original OAI CLIP to filter out low-quality data. Specifically, in the first line of works, HYPE [3] leverages embeddings from hyperbolic models instead of the classical Euclidean-based CLIP to measure how each data point has semantically overlaps with other data points and filters out data with low specificity. T-MARS [12] removes images where the text is the only feature correlated with the caption using FAST [13], an off-the-shelf OCR text detection model. Devil [14] applies fasttext [15] to remove non-English texts and use BLIP-2 [16] model for digit recognition to keep useful images with digits. The second direction, represented by Data Filtering Network (DFN) [2], involves training a new CLIP-style teacher model that uses high-quality datasets like HQITP-350M. Although the embeddings extracted from this model perform worse than the OAI CLIP in downstream tasks, it is particularly good at filtering out low-quality data. Notably, some of these methods can be combined and indeed, merging the selected data from DFN and HYPE achieves current state-of-art as shown in HYPE [3].

Previous works mainly focus on improving the CLIP embedding quality or utilizing an external model to do filtering but employ the CLIP embedding in a suboptimal way by only using classical methods like CLIPScore. In contrast, in this work, we focus on improving the filtering methods themselves for any given CLIP embedding. We show that there are universal and more effective strategies for utilizing any CLIP teacher model, regardless of its architecture (e.g., B/32 or L/14) or the dataset it was trained on (e.g., OpenAI-WIT-400M or DFN's high-quality dataset). These strategies should always be orthogonal to the use of any newly trained CLIP-style models like DFN and might also be compatible with methods using external models like FAST and BLIP-2.

**Our Contributions.** We propose an alternative to CLIPScores that we call **negCLIPLoss** that more accurately characterizes data quality. We also introduce a new distribution metric we call the p-Norm Similarity Score (**NormSim**) when knowledge about downstream tasks is available. Two major observations directly inform our proposals:

- Firstly, we observe that classical methods measure the quality of a multi-modal sample by computing the cosine similarity between its visual and language embeddings, believing that lower similarity indicates that the text does not match its image part well. However, we find that some less informative samples may have a systematic bias, which leads to higher CLIPScores. For example, the language part containing the word "image" can result in higher similarity with any visual part, even when the text does not accurately describe its image content. Our proposed method **negCLIPLoss**, inspired by the standard CLIPLoss, normalizes the original CLIPScore by the similarity between a sample and its contrastive pairs. For example, the high score caused by the word "image" is typically consistent across its contrastive pairs, so our adjustment reduces this bias. As we have highlighted, such replacement can be universally applied across different embedding models. See Fig. 2 for illustrations.
- Secondly, if one has access to examples drawn from the same distribution as the target task, it is natural to assume that this extra knowledge could be leveraged to inform the data filtering process. We propose the **NormSim** metric to measure the vision similarity between a training sample $x$ and the target task dataset $X_{\text{target}}^v \in \mathbb{R}^{n \times D}$ defined as $\|f_v(X_{\text{target}}^v)f_v(x^v)\|_p$, where $f_v : \mathbb{R}^D \to \mathbb{R}^d$ is the vision encoder of teacher model so that $f_v(X_{\text{target}}^v) \in \mathbb{R}^{n \times d}$, $f_v(x^v) \in \mathbb{R}^d$, and $f_v(X_{\text{target}}^v)f_v(x^v) \in \mathbb{R}^n$, and $\|\cdot\|_p$ is the $p$ norm; effective choices are $p = 2$ or $\infty$. Notably, unlike previous ImagetNet-based filtering [1], which tries to keep the training set as diverse as

downstream tasks by clustering the training set and finding the nearest neighbor group for *every target sample*, our method does not explicitly consider the diversity but select examples as long as it is close to *any target sample* (i.e. select high NormSim score). Notably, **negCLIPLoss** and **NormSim** enjoy complementary effect in data selection. See Fig. 3.

To illustrate the effectiveness of our methods, we use a widely used benchmark DataComp [1] as our primary method of evaluating the datasets created by our data filtering methods. We show that, by simply replacing the CLIPScores with **negCLIPLoss** and utilizing **NormSim** we are able to exceed the best OAI-CLIP(L/14)-based baseline by 5.3% on ImageNet-1k and 2.8% on average across 38 downstream tasks, which is similar or even better than the performance achieved by many external-resources-based methods. Notably, even if the target downstream tasks are not available, using NormSim on a proxy downstream task constructed from the training set, called **NormSim$_2$-D**, combined with negCLIPLoss, can also gain a 1.9% improvement on 38 downstream evaluation.

Moreover, the improvements achieved by our methods are not limited to OAI CLIP-based methods but can also be obtained by combining our methods with advanced models that require external resources. *By merging the subset selected by **negCLIPLoss** and **NormSim** with the subset selected by current state-of-the-art method "HYPE ∪ DFN", we can further improve it by 0.9% on both ImageNet-1k and on average 38 downstream tasks. Besides, we can also achieve a 0.8% improvement on average 38 tasks over "HYPE ∪ DFN" using only the data selected by DFN and our strategies.* More importantly, we demonstrate that negCLIPLoss, as a replacement for CLIPScore, can be applied to any other embedding models like OAI-L/14, OAI-B/32, and DFN-B/32, universally boosting performance from 0.4% to 3.0% on an average of 38 tasks. This result is not only technically insightful for understanding the information available in embeddings but also practically significant. Compared to existing methods, our approach saves a significant amount of computational time on both reprocessing and new embedding retraining as shown in Table 5.

## 2    Problem Setup

**Data Filtering on Multimodal Dataset.** We are given a training dataset $D_{\text{train}} = \{x^v, x^l\}$, where $(x^v, x^l) \in \mathbb{R}^D$ is the image-text (vision-language) training pair. For convenience, we will let superscript $vl$ denote either modality so that, for example, $x^{vl} \in x^v, x^l$. Our goal is to identify a subset $S \subset D_{\text{train}}$ that maximizes the zero-shot accuracy of the CLIP model on some downstream tasks when $S$ is used to train the CLIP model.

**CLIP score and embedding.** Recent efforts, such as LAION [5] and DataComp [1], use OpenAI's CLIP ViT-L/14 model [4] as a teacher model to obtain quality score. Here we denote this vanilla CLIP model as $\bar{f}_{vl}$. For any pair $x^{vl}$, the model outputs a normalized unit-vector $\bar{f}_{vl}(x^{vl})$. If $X^{vl} := \{x^{vl}_1, \ldots, x^{vl}_m\}$ denotes a dataset containing $m$ samples, then we define $\bar{f}_{vl}(X^{vl}) = [\bar{f}_{vl}(x^{vl}_1), \ldots, \bar{f}_{vl}(x^{vl}_m)]^\top \in \mathbb{R}^{m \times d}$ as the embedding matrix. The popular filtering metric "CLIPScore" is defined as $\langle \bar{f}_v(x^v), \bar{f}_l(x^l) \rangle \in [-1, 1]$.

**Dataset and model.** Here we follow the pipeline of Datacomp [1] to standardize the training and evaluation process. This is a testbed for dataset experiments aiming to open-source and further improve the vanilla CLIP model and is widely adopted in previous data selection papers [17, 18, 12, 2, 19, 7]. We will give more details in Sec. 4.

## 3    Data Filtering Strategy

### 3.1    negCLIPLoss: A Better Metric than CLIPScore

In this section, we introduce a better and statistically interpretable quality metric called negCLIPLoss, which directly replaces the common metric CLIPScore. Fig. 1 illustrates how negCLIPLoss works. This new metric only requires negligible extra computational costs and no additional external data collection costs. As the name suggested, this metric is inspired by the standard CLIP loss used in the actual training process of the teacher CLIP model, which is defined as

$$\ell_{B^*}(x^{vl}_i) = -\frac{1}{2} \left[ \log \frac{\exp(\bar{f}_v(x^v_i)^\top \bar{f}_l(x^l_i)/\tau)}{\sum_{j \in B^*} \exp(\bar{f}_v(x^v_i)^\top \bar{f}_l(x^l_j)/\tau)} + \log \frac{\exp(\bar{f}_v(x^v_i)^\top \bar{f}_l(x^l_i))/\tau}{\sum_{j \in B^*} \exp(\bar{f}_v(x^v_j)^\top \bar{f}_l(x^l_i)/\tau)} \right] \quad (1)$$

Here $B^*$ is the random batch where $i$-th sample belongs during a particular training step, and $\tau$ is the learnable temperate parameter. Notably, the teacher loss differs from CLIPScore primarily by a

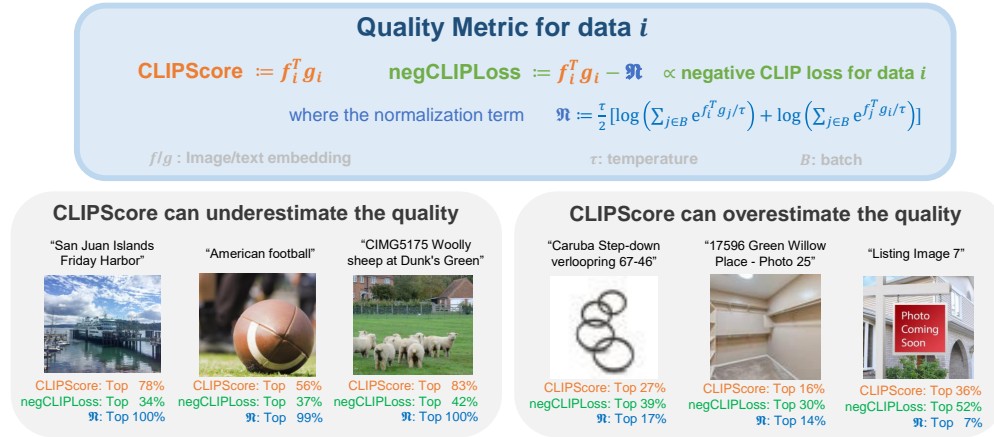

Figure 1: Illustration of negCLIPLoss. CLIPScore may underestimate (bottom left, where the data quality is high but CLIPScore is low) or overestimate (bottom right, where the data quality is low but CLIPScore is high) the quality of image-text pairs. However, this issue can be mitigated by simply subtracting a normalization term $\mathcal{R}$. negCLIPLoss employs the teacher model to calculate the negative CLIP loss on training data and serves as a more accurate metric. Here, "Top X%" denotes that the score represents the top X% *high* values within the entire dataset (i.e., the (100-X)% percentile among all the values). For example, "$\mathcal{R}$ : Top 100%" means this data has almost the smallest $\mathcal{R}$ among the whole dataset, which represents that it contains highly specific elements in both images and texts.

normalization term $\mathcal{R}^*$ as follows:

$$-\tau \cdot \ell_{B^*}(x_i^{vl}) = \underbrace{\bar{f}_v(x_i^v)^\top \bar{f}_l(x_i^l)}_{\text{CLIPScore}} - \underbrace{\frac{\tau}{2} \left[ \log \sum_{j \in B^*} \exp(\frac{\bar{f}_v(x_i^v)^\top \bar{f}_l(x_j^l)}{\tau}) + \log \sum_{j \in B^*} \exp(\frac{\bar{f}_v(x_j^v)^\top \bar{f}_l(x_i^l)}{\tau}) \right]}_{\text{normalization term } \mathcal{R}^*}$$

In practice, since the training dataset of teacher CLIP models, like OAI-WIT400M [4], and the actual batch divisions $B^*$ is inaccessible, we randomly select $K$ batches from the student model's training data and use the averaged results from $\{B_k\}_{i=1}^K$ to estimate the normalization term $\mathcal{R}^*$ on $B^*$:

$$\text{negCLIPLoss}(x_i^{vl}) := -\frac{\tau}{K} \sum_{k=1}^K \ell_{B_k}(x_i^{vl}) \approx \text{CLIPScore}(x_i^{vl}) - \mathcal{R}^* \qquad (2)$$

Here $\{B_k\}_{i=1}^K$ are some batches randomly selected from the student model's training data and $x_i \in B_k, \forall k$. We choose $K = 10$ in our experiments, but any sample size larger than 5 is sufficiently stable for estimating the original CLIPLoss (Details in Appendix D.1). Besides, in Sec. 4.3.3 we also show that the computational cost introduced by $\mathcal{R}$ remains negligible compared to other baselines. The temperature $\tau$ and batch size $|B^*|$ can be directly obtained from the parameters of the pretrained teacher model. More details of negCLIPLoss are in Appendix, including the concentration analysis of $\mathcal{R}$ (Appendix A.1), pseudocode (Algorithm 1), and the ablation study of $\tau$ and $|B|$ (Appendix C.2).

**Motivation behind negCLIPLoss.** Other existing works also use loss-guided data selection, such as LESS [20] in NLP, CoDis [21] in CV, and RHO [22] in general data scheduling scenarios. However, it is still unclear whether selecting based on teacher loss is suitable for multi-modal contrastive learning. Here we give an affirmative answer as shown in Fig. 2, where we can see negCLIPLoss performs better than or on par with CLIPScore consistently.

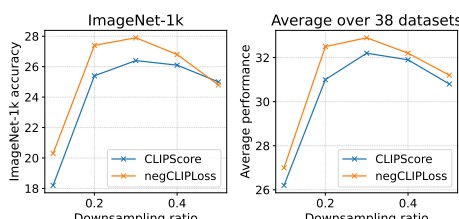

Figure 2: Comparison of negCLIPLoss and CLIPScore across different downsampling ratios on DataComp-medium.

To illustrate how teacher loss helps our selection, we demonstrate that the normalization term provided by negCLIPLoss is crucial for correcting the overestimation or underestimation inherent in CLIPScore. A high normalization term implies that either the image embedding, text embedding, or both can easily match multiple contrastive pairs beyond their

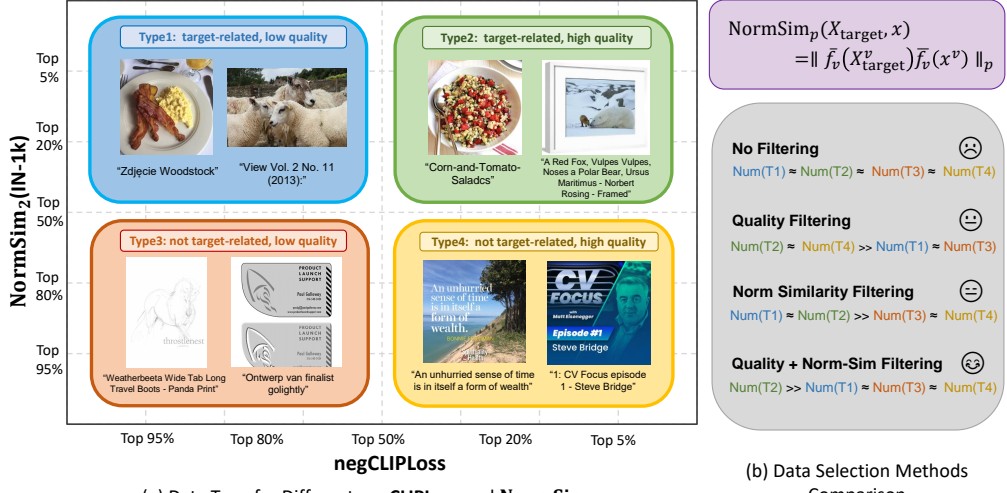

(a) Data Type for Different **negCLIPLoss** and **NormSim$_2$**

(b) Data Selection Methods Comparison

Figure 3: Illustration of NormSim on DataComp. $X_{\text{target}}$ is the target prior data. "Top X%" denotes that the score represents the top X% high values within the entire dataset. (a) Visualization of data with different NormSim and negCLIPLoss. Here we use NormSim$_2$(ImageNet-1k) as an example. Although both Type 2 and Type 4 data have high negCLIPLoss and thus high quality, data with low NormSim$_2$ (Type 4) are more irrelevant to downstream tasks like ImageNet, VTAB, and MSCOCO. For example, they contain many images dominated by OCR content and make little contribution to improving downstream performance. (b) Illustration of a rough comparison of sampling data for different filtering methods. Using "negCLIPLoss $\cap$ NormSim" filtering can balance the quality and relevance to downstream tasks, thus increasing the proportion of Type 2 data. (Refer to Appendix E for more visualization.)

corresponding counterparts. For example, in the bottom right of Fig. 1, the text containing "Image" or "Photo" can be easily matched with any visual content. Similarly, the image of "verloopring" only contains very simple features and can be matched with many words like "white", "empty" or "circle", etc. Consequently, despite a high absolute CLIPScore, the relative negCLIPLoss within its batch can be lower. In contrast, the bottom left features highly specific elements in both text and images, such as "Islands Harbor," "American football", and "sheep at green". These elements are specific and less likely to match with contrastive pairs, resulting in a higher relative negCLIPLoss.

## 3.2 NormSim: A New Training-Target Similarity Metric

Our proposed negCLIPLoss is a universal approach to improve filtering performance by estimating quality better, and it does not rely on any downstream task. Now, if we can access some knowledge of the downstream tasks, we could further improve the performance by using a vision-only *p-norm similarity to target data* metric to measure the relationship between each training sample and the downstream target data. We will discuss the reason to use vision-only embedding later in this section.

Specifically, we assume access to the target set of downstream tasks and denote them as $X_{\text{target}} = \{x_{\text{target},(1)}, \ldots, x_{\text{target},(m)}\}$, where each $x_{\text{target},(i)} \in \mathbb{R}^d$ is *i.i.d.*-sampled from the target downstream distribution $\mathcal{P}_{\text{target}}$[3], but without overlapping with the test set. Then, for each training sample $x^{vl}$ and the corresponding target set $X_{\text{target}}$, the NormSim is defined as:

$$\text{NormSim}_p(X_{\text{target}}, x) := \|\bar{f}_v(X^v_{\text{target}})\bar{f}_v(x^v)\|_p = \left( \sum_{x_t \in X_{\text{target}}} \left| \langle \bar{f}_v(x^v_t), \bar{f}_v(x^v) \rangle \right|^p \right)^{1/p} \quad (3)$$

We select the subset $S$ by choosing the samples with top-$N$ highest NormSim scores. The choice of the norm type $p$ can be based on the data distribution and training process. In this paper, we consider two instantiations of $p$:

When $p = 2$, our data selection method can be regarded as the following equation. It's equivalent to selecting a subset that aligns with the principal components of the target set variance (Appendix C.6.1).

---

[3]Although out-of-distribution tasks like "WILDS" have distribution shift between training data and test data, they still provides useful information of the test data.

$$S = \arg\max_{|S|=N} \sum_{i \in S} \mathrm{NormSim}_2(x_t, x_i), \quad \mathrm{NormSim}_2(x_t, x_i) = \left( \sum_{x_t \in X_{\mathrm{target}}} \left| \bar{f}_v(x_t^v)^\top \bar{f}_v(x^v) \right|^2 \right)^{1/2} \tag{4}$$

When $p = \infty$, the distance metric can be regarded as an even more optimistic measure, such that a training sample will be selected if it has high similarity to *any target sample*. Note that this is different from nearest-neighbor-based method used in image-based filtering [1], where they are trying to find the nearest training sample of *every target sample*. In this case, it can be regarded as:

$$S = \arg\max_{|S|=N} \sum_{i \in S} \mathrm{NormSim}_\infty(x_t, x_i), \qquad \mathrm{NormSim}_\infty(x_t, x_i) = \max_{x_t \in X_{\mathrm{target}}} \bar{f}_v(x_t^v)^\top \bar{f}_v(x_i^v) \tag{5}$$

In Appendix D.3, we also show that our $\mathrm{NormSim}_\infty$ can outperform the nearest neighbor selection on the downstream target tasks. Here, we show an example selected via the $\mathrm{NormSim}_2$(ImageNet-1k) in Fig. 3, showing that this vision-target-aware method is complementary to the quality-based one.

**Choice of Target Data.** In the experiment parts, we try two kinds of target data: training data from ImageNet-1k (1.3M) or training data from all 24 accessible downstream tasks (2.1M)[4]. We denote them as **NormSim$_p$(IN-1k)** and **NormSim$_p$(Target)**, respectively.

**Necessity of using vision-only information** We use only the visual information $x^v$ instead of multi-modal information $x^{vl}$ for measuring similarity. This is because common crawled text often has brief captions, making the OAI CLIP language embedding weaker than its visual embedding model [1, 23–25]. Consequently, the language part cannot characterize the pre-training and downstream task distribution as well as the visual part. This phenomenon is also observed in Gadre et al. [1], where image-based filtering (select data whose image embeddings are similar to that from ImageNet-1k) outperforms text-based filtering (select data whose captions contain words from ImageNet-21k). More ablation studies are provided in Appendix D.4.

**Generality of NormSim in choosing teacher model.** Notably, since we just use image embeddings in the NormSim metric, we believe it unnecessary to use CLIP model to obtain NormSim. Norm-Sim can be a general metric for selecting target-related image/image-text data if any good image representations are given, like the representations obtained from pretrained ResNet-50.

**Theoretical justification.** Unlike many existing methods that force diversity by selecting training samples around each $x_{\mathrm{target}}$, our strategy maximizes similarity without directly considering data diversity. For the $p = 2$ case, we demonstrate that maximizing $\mathrm{NormSim}_2$ is optimal under a linear model $\bar{f}_v$, as shown in Appendix A.2. Our theorem also provides error guarantees for noisy embeddings and explains when vision-only embeddings outperform combined vision and language embeddings. Recent work by Joshi et al. [26] provides a similar analysis but focuses on high-quality data and cross-variance between images and texts. This approach is less effective than image-only methods for filtering noisy datasets, as discussed above.

**Using proxy when downstream $X_{\mathrm{target}}$ is inaccessible.** Surprisingly, we show that the 2-norm can also be used when only the pre-training set is available. In this case, we construct a proxy "target" set from the pre-training set itself. Specifically, let $S_i$ be the selected subset at step $i$, then we treat the current $S_i$ as the proxy "target" set. To construct the next smaller set, we select the next data batch $S_{i+1}$ satisfying $\arg\max_{S_{i+1} \subset S_i} \sum_{x \in S} \mathrm{NormSim}_2(S_i, x)$, until reaching an N size subset. We call this approach **NormSim$_2$-D** (Dynamic) and will specify the algorithm details in Appendix C.3.

## 4 Experimental Results

In this section, we evaluate the performance of negCLIPLoss and NormSim, aiming to address the following questions: **Q1:** Given a fixed CLIP teacher model, can our methods more effectively utilize CLIP embeddings for data filtering? **Q2:** Are our methods applicable to diverse CLIP teacher models with varying architectures or different pretrained datasets? **Q3:** How does our method compare to other leading approaches that utilize external models or multimodal datasets? Additionally, could our method be compatible with these methods and enhance their effectiveness?

---

[4]Here we only use the target data for data selection, instead of training on them. The target dataset is significantly smaller than pretraining set like DataComp-medium (128M) or external datasets like HQITP-350M utilized by DFN [2].

## 4.1 Setup

We adhere to the standardized training and evaluation protocols of the DataComp benchmark [1]. **Training configuration.** We employ the medium-scale training configuration of DataComp (DataComp-medium). It provides a substantial dataset comprising 128 million low-quality, web-curated image-text pairs to be filtered. Once the data subset is obtained by some data filtering strategy, it will be used to train a fixed CLIP-B/32 model in a fixed training budget that allows the model to pass 128 million data points an epoch. Therefore, smaller subsets will be repeated more frequently, ensuring a fair comparison. We note that the size of the DataComp dataset becomes smaller over time since some URLs of images become invalid[5], and we only successfully downloaded about 110M data. Therefore, the results of baselines on the leaderboard do not apply to our datasets, and we reproduce all the top baselines on the leaderboard with their public UIDs of the selected data.

**Evaluation.** We measured the model performance on 38 downstream datasets including image classification and retrieval tasks followed by DataComp. The image classification tasks contain ImageNet-1k [27], ImageNet distribution shifts [28–31], 11 datasets from the Visual Task Adaptation Benchmark (VTAB) [32] and 3 datasets from WILDS [33, 34]. Retrieval datasets contain Flickr30k [35], MSCOCO [36] and WinoGAViL [37].

**Teacher model architecture.** Our experiments utilize two architectures for OpenAI's CLIP teacher models: ViT-L/14 and ViT-B/32. Additionally, we use the public version of DFN (DFN-P) proposed by Fang et al. [2] as a teacher model, and its architecture is also ViT-B/32.

## 4.2 Baselines

We restate the three current research directions mentioned before based on how much external resources are employed: (D1) using OAI CLIP alone while optimizing embedding employment strategies, (D2) training and using a more advanced CLIP embedding model based on external data, and (D3) utilizing non-CLIP external models to aid data selection. It is important to note that D2 and D3 may also incorporate strategies from D1. For example, CLIPScore (D1) has been used in almost all the top methods. Therefore, we categorize baselines by the largest possible category they encompass. According to the above categorization, we summarize the baselines we used in our experiments as follows. Please refer to Fig. 4 and Appendix C.4 for more details.

**D1: OAI CLIP embedding only.** The learner can only access the pretraining dataset (like DataComp-medium), the original OAI CLIP teacher model that is used to extract embeddings, and some target data of the downstream tasks which is much smaller than the pretraining dataset (like ImageNet-1k). In this category, we don't use any existing external non-CLIP models or any newly trained CLIP model based on external multi-modal dataset. In detail, This category includes (1) **CLIPScore** [38], which only uses CLIPScore for filtering as we mentioned before. (2) **Image-based filtering** [1], which uses ImageNet-1K training data as the downstream target data for data filtering. It applies k-means clustering to the *image* embeddings of training data and selects clusters closest to the ImageNet-1K embeddings. Gadre et al. [1] also try to combine image-based filtering and CLIPScore together. (3) $\mathbb{D}^2$ **Pruning** [18], which represents the dataset as an undirected graph and selects the data by combining difficulty and diversity. They use the CLIP score to initialize their graph.

**D2, D3: Accessible external model and multi-modal data.** All the current top baselines enable the learner to utilize external resources, either to train a better CLIP teacher model or to help filtering using existing models' properies. In detail, (1) **DFN** [2] trains another CLIP data filtering network via external high-quality data. Their currently public model (**DFN-P**) is trained on CC12M [39] + CC3M [40] + SS15M [41], while the best DFN is trained on nonpublic HQITP-350M [2], which is even larger than DataComp-medium. (2) **HYPE** [3] leverages hyperbolic embeddings (different from CLIP embedding) and the concept of entailment cones to filter out samples with meaningless or underspecified semantics, enhancing the specificity of each sample. (3) **HYPE ∪ DFN** proposed by [3] samples subset separately for each method and then merge them. This is the state-of-the-art method on the DataComp benchmark for medium size. (4) Other methods including **T-MARS** [12], **Devils** [14], **MLM** [42], which leverage external models such as text detection model FAST [13], BLIP-2 [16] and LLaVA-1.5 [43, 44] to heuristically select data. See details in Appendix C.4.

**Cross-setting comparison.** We make these separations for fair comparison. Intuitively, performance should be ranked as **D2, D3 > D1**. However, our results show that cross-setting comparisons are possible and our D1 methods can perform similar or even better than most of D3 methods.

---

[5]See https://github.com/mlfoundations/datacomp/issues/3. Similar issues are proposed by $\mathbb{D}^2$ pruning [18].

Table 2: Results on DataComp-medium from methods that use only OpenAI's CLIP-L/14 model, i.e., all methods are from the **D1** category. The "dataset size" represents the size of the subset obtained from different approaches. NormSim(IN-1k) denotes using the training data of ImageNet-1k as the target while NormSim(Target) represents using that of all 24 available downstream tasks. NormSim-D refers to the methods that use an iteratively selected subset from the training set as the target proxy.

| Filtering Strategy | Dataset Size | IN-1k (1 task) | IN Dist. Shift (5) | VTAB (11) | Retrieval (3) | Avg. (38) |
|---|---|---|---|---|---|---|
| No filtering [1] | 110M | 17.3 | 15.0 | 25.2 | 21.3 | 25.6 |
| CLIPScore (20%) [38] | 22M | 25.4 | 22.7 | 31.8 | 22.0 | 31.0 |
| CLIPScore (30%) [38] | 33M | 26.4 | 23.6 | 32.6 | 24.5 | 32.2 |
| Image-based [1] | 24M | 25.5 | 21.9 | 30.4 | 24.6 | 29.9 |
| CLIPScore (30%) ∩ Image-based [1] | 11M | 27.4 | 23.9 | 31.9 | 21.4 | 30.8 |
| $\mathbb{D}^2$ Pruning [18] | 22M | 23.2 | 20.4 | 31.4 | 18.7 | 29.5 |
| negCLIPLoss (20%) | 22M | 27.4 | 23.8 | 33.7 | 23.7 | 32.5 |
| negCLIPLoss (30%) | 33M | 27.9 | 24.6 | 33.2 | 25.1 | 32.9 |
| CLIPScore (30%) ∩ NormSim$_2$-D | 22M | 28.3 | 25.0 | 34.5 | 22.7 | 32.9 |
| negCLIPLoss (30%) ∩ NormSim$_2$-D | 22M | 29.8 | 26.1 | 34.8 | 24.6 | 34.1 |
| CLIPScore (30%) ∩ NormSim$_2$(IN-1k) | 22M | 29.1 | 25.4 | _35.8_ | 24.1 | 33.4 |
| CLIPScore (30%) ∩ NormSim$_2$(Target) | 22M | 28.9 | 25.1 | 32.7 | 23.6 | 32.5 |
| CLIPScore (30%) ∩ NormSim$_\infty$(IN-1k) | 22M | 29.7 | 25.9 | 33.7 | 24.1 | 33.7 |
| CLIPScore (30%) ∩ NormSim$_\infty$(Target) | 22M | 30.2 | 26.2 | 35.0 | 23.4 | 33.9 |
| negCLIPLoss (30%) ∩ NormSim$_2$(IN-1k) | 22M | 30.4 | 26.4 | 35.4 | _25.6_ | 34.3 |
| negCLIPLoss (30%) ∩ NormSim$_2$(Target) | 22M | 30.6 | 26.2 | 35.2 | 25.5 | 33.9 |
| negCLIPLoss (30%) ∩ NormSim$_\infty$(IN-1k) | 22M | **31.9** | **27.3** | 34.8 | 25.0 | _34.4_ |
| negCLIPLoss (30%) ∩ NormSim$_\infty$(Target) | 22M | _31.7_ | _27.2_ | **36.0** | **26.0** | **35.0** |

## 4.3 Main Results and Discussions

### 4.3.1 Comparision on D1 Category (Q1)

In Table 2, we compare the D1 methods where only the OAI CLIP model is allowed to be used.

**Our Methods leverage OAI CLIP-L/14 better.** *First*, negCLIPLoss outperforms CLIPScore on *all metrics*, regardless of whether it is used alone or combined with other methods. These results support our claim that negCLIPLoss can more accurately estimate the data quality.

*Second*, even when target knowledge is unavailable, use NormSim$_2$-D together with negCLIPLoss can still improve the filtering performance by 1.9% on average 38 downstream tasks. *Third*, when target knowledge is available, NormSim$_2$ and NormSim$_\infty$ can improve filtering more significantly compared with NormSim$_2$-D, and *in general, NormSim$_\infty$ is the best choice*. Especially, compared with the best baseline 'CLIPScore (30%)', our best combination 'negCLIPLoss ∩ NormSim$_\infty$(Target)' improves **5.3%** on **ImageNet-1k** and **2.8%** on average **38 downstream tasks**, respectively. Later in Table 3 we will see that this result outperform all the D3 baselines except DFN ∪ HYPE. On the other hand, when using ImageNet-1k as the target data, the choice of norm has very little influence.

Table 1: Results on DataComp-medium from the top methods that use only OpenAI's CLIP-B/32 model or public version of DFN (DFN-P). "NormSim$_\infty^{B/32}$" represents using OAI CLIP-B/32 to calculate NormSim$_\infty$.

| Strategy | Size | IN-1k | VTAB | Avg. |
|---|---|---|---|---|
| **OAI CLIP-B/32** | | | | |
| CLIPScore (30%) | 33M | 27.6 | 33.6 | 33.2 |
| CLIPScore (20%) | 22M | 27.0 | 33.0 | 32.2 |
| negCLIPLoss (30%) | 33M | 28.8 | 33.7 | 33.6 |
| negCLIPLoss (20%) | 22M | 28.9 | 34.3 | 33.0 |
| negCLIPLoss (30%) ∩ NormSim$_\infty$(Target) | 22M | **32.4** | **35.9** | **35.2** |
| **DFN-P** | | | | |
| CLIPScore (30%) | 33M | 28.4 | 33.2 | 32.7 |
| CLIPScore (20%) | 22M | 29.7 | 33.0 | 33.1 |
| CLIPScore (17.5%) | 19M | 30.2 | 34.1 | 33.8 |
| CLIPScore (15%) | 16M | 25.9 | 32.9 | 31.6 |
| negCLIPLoss (30%) | 33M | 28.9 | 33.4 | 33.2 |
| negCLIPLoss (20%) | 22M | 30.7 | 33.6 | 33.8 |
| negCLIPLoss (17.5%) | 19M | 31.2 | 35.7 | _34.7_ |
| negCLIPLoss (15%) | 16M | 31.3 | _35.8_ | 34.6 |
| negCLIPLoss (30%) ∩ NormSim$_\infty$(Target) | 22M | 29.4 | 33.5 | 32.5 |
| negCLIPLoss (17.5%) ∩ NormSim$_\infty$(Target) | 16M | _31.5_ | 34.6 | 34.4 |
| negCLIPLoss (17.5%) ∩ NormSim$_\infty^{B/32}$(Target) | 16M | **31.6** | **37.2** | **35.7** |

Table 3: Results of all D1&D2&D3 top methods on DataComp-medium. The results of MLM [42] are from their paper, while all other baselines are reproduced on our downloaded dataset using their official UIDs. "Ours (20%)" refers to use "negCLIPLoss (30%) ∩ NormSim$_\infty$(Target)" to get 20% of original data, while "Ours (10%)" denotes applying "negCLIPLoss (20%) ∩ NormSim$_\infty$(Target)" to get 10%. And we use "*" to indicate the case where we choose the intersection of the data selected by using OAI CLIP-B/32 and OAI CLIP-L/14 separately, which results in about 15M data for "Ours (20%)*" and 7.4M data for "Ours (10%)*".

| Type | Filtering Strategy | Dataset Size | IN-1k (1) | IN Dist. Shift (5) | VTAB (11) | Retrieval (3) | Avg. (38) |
|------|-------------------|--------------|-----------|--------------------|-----------|---------------|-----------|
| D3 | T-MARS [12] | 22M | 30.8 | 26.3 | 34.8 | 25.4 | 34.1 |
| D3 | Devil [14] | 20M | 31.0 | 26.7 | 35.9 | 24.7 | 34.5 |
| D3 | MLM [42] | 38M | 30.3 | 25.6 | 36.0 | **29.0** | 34.5 |
| D3 | HYPE [3] | 10M | 30.3 | 25.8 | 34.3 | 22.2 | 31.9 |
| D2 | DFN [2] | 16M | 36.0 | 30.1 | 36.2 | 27.0 | 35.4 |
| D3 | DFN ∪ HYPE [3] | 20M | 36.4 | 30.8 | 38.5 | 28.0 | 36.8 |
| D1 | **Ours (20%)** | 22M | 32.4 | 27.4 | 35.9 | 26.3 | 35.2 |
| D3 | DFN ∪ **Ours (20%)*** | 23M | 36.4 | 30.9 | **38.6** | 28.1 | 37.6 |
| D3 | DFN ∪ HYPE ∪ **Ours (10%)*** | 22M | **37.3** | **31.4** | 38.5 | 27.6 | **37.7** |

#### 4.3.2 Try Other Teacher Models (Q2)

To evaluate whether our method applies to other CLIP teacher models, we replaced OAI CLIP-L/14 with OAI CLIP-B/32 and DFN-P as embedding models. We compare the best baseline "CLIPScore" with our "negCLIPLoss" and best strategy "negCLIPLoss ∩ NormSim$_\infty$(Target)" as shown in Table 1 and Appendix D.2. Note that the original DFN paper selects a subset comprising 19.2M data points, which accounts for approximately 17.5% of our dataset and 15% of their dataset, we incorporate these sampling ratios into our comparison.

**negCLIPLoss can be applied to different CLIP embedding models.** Our proposed negCLIPLoss, as a replacement of CLIPScore, not only leads to better performance compared to all the other baselines using OAI CLIP-L/14 as shown in Table 2, but also achieves universal improvement on the other two CLIP embedding models, OAI CLIP-B/32 and DFN-P as shown in Table 1. Our methods can consistently outperform all downstream tasks for different filtering ratios and models, like a 0.5%-5.4% increase on ImageNet-1k.

**Embedding required by NormSim should have good downstream performance.** When combining negCLIPLoss with NormSim$_\infty$, OAI CLIP-B/32 and DFN-P exhibit completely different behaviors. The former obtains results even better than those in Table 2, which uses OAI CLIP-L/14 as the teacher model, while DFN-P achieves results even worse than using negCLIPLoss alone[6]. The reason is that, unlike OAI CLIP-B/32, DFN-P is specially designed for data filtering *at the expense of downstream task performance*, as claimed by its authors. For example, the ImageNet-1k accuracy for DFN-P, OAI CLIP-B/32, and OAI CLIP-L/14 are 45%, 63%, and 75%, respectively. This indicates that the embeddings obtained from DFN on target data might be highly unreliable, leading to inaccurate similarity calculations between training and target data. To support this, if we use DFN-P to evaluate negCLIPLoss but utilize OAI CLIP-B/32 for calculating NormSim, as shown in "negCLIPLoss (17.5%) ∩ NormSim$_\infty^{B/32}$(Target)", we can further improve the results compared to using negCLIPLoss alone. Its average performance on 38 tasks is even higher than utilizing the best DFN (trained on HQITP-350M) with CLIPScore, as shown in Table 3.

#### 4.3.3 Comparison with D2 & D3 Categories (Q3)

In this part, we compare all the D2 & D3 baselines mentioned in Sec. 4.2 together with our best strategy in Table 3. Here we reproduce all the baselines if their official UIDs are available. For "A ∪ B" mentioned in Table 3, we follow the way of "HYPE ∪ DFN" in Kim et al. [3] to merge the data, which generates the sampling subset separately for each method and then merge them. This will result in oversampling the shared data, which is intuitively more important.[7] We also show the best result

---

[6]see "negCLIPLoss (30%) ∩ NormSim$_\infty$(Target)" versus "negCLIPLoss (20%)/(30%)" and "negCLIPLoss (17.5%) ∩ NormSim$_\infty$(Target)" versus "negCLIPLoss (17.5%)/(15%)"

[7]For the dataset size of "A∪B", we count the number of the unique data in the dataset followed HYPE [3].

we obtain by combining our method with DFN [2] and HYPE [3] on the full DataComp-medium dataset in Table 4, where the baselines are from DataComp benchmark.

**Our methods can outperform most of the D3 methods.** In Table 3, we show that without using any external models or data, our best combination, i.e., using OAI CLIP-B/32 for "negCLIPLoss (30%) ∩ NormSim$_\infty$(Target)" (**Ours (20%)**), still outperforms all methods except DFN and "DFN ∪ HYPE". This answers the first part of Q3 and further indicates that some external models may be redundant since CLIP embeddings already contain necessary information.

**We can further improve the SOTA method.** In Table 3, we show that our model can further boost the performance of the current SOTA method "HYPE ∪ DFN" by 0.9% on both ImageNet-1k and

Table 4: Results of the top methods on the full DataComp-medium dataset (128M data).

| Strategy | IN-1k | Avg. |
|---|---|---|
| No filtering | 17.6 | 25.8 |
| CLIPScore [38] | 27.3 | 32.8 |
| T-MARS [12] | 33.0 | 36.1 |
| Devils [14] | 32.0 | 37.1 |
| DFN [2] | 37.1 | 37.3 |
| DFN ∪ HYPE [3] | **38.2** | 37.9 |
| DFN ∪ **Ours (20%)** | 37.5 | 38.6 |
| DFN ∪ HYPE ∪ **Ours (10%)** | **38.2** | **38.8** |

on average 38 downstream tasks, and close results can be achieved even without combining HYPE which utilizes the external embedding model MERU [45]. And we update the SOTA performance of the DataComp-medium (full dataset) benchmark as shown in Table 4. Here we use the data selected by both OAI CLIP-B/32 and L/14, which we found is more robust than using one of them alone. Our better results answer the second part of Q3, that is, our methods can be compatible with other D2&D3 methods.

## 5   Conclusion and Limitation

In this paper, we introduce two metrics, negCLIPLoss and NormSim, to enhance data selection in multimodal contrastive learning without relying on external resources. negCLIPLoss provides a more accurate quality metric compared to the commonly used CLIPScore, while NormSim measures the similarity between pretraining data and target data for known downstream tasks. Experiments show that our methods achieve results that are competitive with or even better to approaches using external models or datasets. Additionally, negCLIPLoss and NormSim are compatible with existing top techniques, allowing us to achieve a new state-of-the-art by combining them.

A notable limitation of our study is the exclusion of larger pretraining datasets, such as the large and xlarge scales of DataComp. However, DataComp-medium is the most commonly used benchmark for data selection in CLIP pretraining, and our method has demonstrated both effectiveness (Table 2-3) and efficiency (Table 5) on it. Future directions include exploring better ways to merge data selected by different methods and incorporating our methods into data scheduling scenarios.

## 6   Acknowledgement

We thank Tong Chen, Pang Wei Koh, Xiaochuang Han, Rui Xin, Luyao Ma, Lei Chen, and other members in the UW ML Group for many insightful discussions and helpful feedback. The research of Kevin Jamieson and Yifang Chen are partially supported by the NSF through the University of Washington Materials Research Science and Engineering Center, DMR-2308979, and awards CCF 2007036. SSD acknowledges the support of NSF IIS 2110170, NSF DMS 2134106, NSF CCF 2212261, NSF IIS 2143493, NSF CCF 2019844, and NSF IIS 2229881.

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

# A Theoretical Interpretation

## A.1 Concentration of Normalization Term in negCLIPLoss

In this section, we construct a theorem using the concentration inequality to show that when the batch size is sufficiently large, the normalization term $R^{B_k}$ obtained from actual batch $B_k$ can approximate $R^{B^*}$ calculated using ground truth batch $B^*$ quite well. The details are as follows:

We assume that the pretraining dataset $\mathcal{D}$ is ndependent and identically distributed (*i.i.d.*) sampled from some distribution $\mathcal{P}$. Besides, to use pretraining data batch to approximate the ground truth batch, one necessary condition is that their distribution is similar. Here for simplicity, we assume that they are also *i.i.d.*.

**Assumption A.1.** *We assume that the ground-truth batch of data $B^*$ used by the teacher model is i.i.d. to the pretraining dataset $\mathcal{D}$ which is required to be filtered.*

For simplicity, we denote $s_{ij} = \bar{f}_v(x_i^v)^\top \bar{f}_l(x_j^l), i, j \in B$ to be the cross-image-text similarities in the batch $B$. Then the normalization term can be written as

$$\mathcal{R}_i^B = \frac{\tau}{2}\left[\log(\sum_{j\in B}\exp(s_{ij}/\tau)) + \log(\sum_{j\in B}\exp(s_{ji}/\tau))\right]$$

Here $s_{ij} \in [-1, 1]$. We will show that $\mathcal{R}_i^B = (1 + o(1)) \cdot \mathcal{R}_i^{B^*}$ for all $i$ when $|B|$ is sufficiently large, which means that we can use the random batch to approximate the ground-truth batch.

**Theorem A.1.** *If Assumption A.1 holds and the batch size satisfies $|B| = |B^*|$, then we have $\mathcal{R}_i^B = \Theta(\log(|B|))$ while $|\mathcal{R}_i^B - \mathcal{R}_i^{B^*}| = O(\frac{1}{\sqrt{|B|}})$ for any $i \in B \cap B^*$.*

*Proof.* Since $s_{ij} \in [-1, 1]$, It's obvious that $\mathcal{R}_i^B = \Theta(\log(|B|))$. Let $\alpha_{ij} := \exp(s_{ij}/\tau) - \mathbb{E}_j[\exp(s_{ij}/\tau)]$, then $\alpha_{ij}$ is zero-mean. Note that since the data is *i.i.d.*, so does $\alpha_{ij}$, and we denote $\gamma := \mathbb{E}_j[\alpha_{ij}^2]$. Note that $|\alpha_{ij}| \leq e^{1/\tau} =: M$, from Bernstein inequality we have

$$\mathbb{P}(|\sum_{j\in B}\alpha_{ij}| \geq t) \leq 2\exp(-\frac{\frac{1}{2}t^2}{|B|\gamma + \frac{1}{3}Mt})$$

A similar conclusion holds for $B^*$. These result that with probability at least $1 - \eta$, we have

$$|\sum_{j\in B}\alpha_{ij}| \leq \max\{2\sqrt{|B|\gamma\ln(\frac{2}{\eta})}, \frac{4}{3}M\ln(\frac{2}{\eta})\} =: t(|B|, \gamma, \eta, M)$$

Thus we have $|\sum_{j\in B}\exp(\frac{s_{ij}}{\tau}) - \sum_{j\in B^*}\exp(\frac{s_{ij}}{\tau})| \leq 2t(|B|, \gamma, \eta)$. Furthermore, for any $x_1, x_2 > 1$, it's easy to prove that $|\log(x_1) - \log(x_2)| \leq \frac{|x_1-x_2|}{\min(x_1,x_2)}$. Therefore, we have $|\log(\sum_{j\in B}\exp(\frac{s_{ij}}{\tau})) - \log(\sum_{j\in B^*}\exp(\frac{s_{ij}}{\tau}))| \lesssim O(\frac{1}{\sqrt{|B|}})$. Similar claims hold for $|\mathcal{R}_i^B - \mathcal{R}_i^{B^*}|$.

$\square$

## A.2 Optimality of NormSim$_2$ Under Linear Assumption

In this section, we give a theoretical justification on the NormSim metric when $p = 2$ under the linear model assumptions when low quality image and mismatched text has already been removed. In other words, we mainly focus on the following strategy.

$$S = \arg\max_{|S|=N}\sum_{i\in S}\bar{f}_v(x_i^v)^\top \underbrace{\left(\frac{1}{|X_{\text{target}}|}\sum_{x_t\in X_{\text{target}}}\bar{f}_v(x_t^v)\bar{f}_v(x_t^v)^\top\right)}_{\bar{\Sigma}_{\text{target\_proxy}}}\bar{f}_v(x_i^v) \tag{6}$$

### A.2.1 Theoretical Setup

**Training data.** For any $\boldsymbol{x}^v, \boldsymbol{x}^l \in \mathbb{R}^d$ observable image and text training pairs, we define $\boldsymbol{z}^v, \boldsymbol{z}^l$ to be the corresponding latent vectors which contain all semantically pertinent information about our tasks of interest. Similar to previous theoretical work [46], we assume each i.i.d pair $\boldsymbol{z}^{vl}$ follows zero-mean sub-gaussian distribution whose cross-covariance satisfies

$$\text{Cov}(\boldsymbol{z}^v, \boldsymbol{z}^l) = \Sigma_{\text{train}} = \text{diag}(\sigma_1, \sigma_2, \dots), \qquad \|\boldsymbol{z}^{vl}\| = 1$$

and each $\boldsymbol{x}^{vl}$ is generated based on a linear model such that

$$\boldsymbol{x}^{vl} = G_{vl}^* \boldsymbol{z}^{vl} + \boldsymbol{\xi}^{vl}.$$

Here $G_{vl}^* \in O_{d \times r}$ is the othonormal ground truth representation mapping from the latent vector space to the input space, and $\xi^{vl} \sim \mathcal{N}(0, I_d)$ are *i.i.d.* random noise.

Also we denote the cross covariance of any finite dataset $S'$ (e.g. the given train set $D_{\text{train}}$) as $\Sigma_{S'}$.

**Test data.** For any zero-shot downstream task, we assume it shares almost same data generation process as the training set, except its the cross-covariance $\Sigma_{\text{target}}$ does not necessarily equal $\Sigma_{\text{train}}$, which necessitate the choice of $\bar{\Sigma}_{\text{target\_proxy}}$.

**CLIP embedding model as teacher.** Under the linear model assumption, we have a teacher model $\bar{f}_{vl} = \bar{G}_{vl}$, whose generated clip embedding can partially recover the ground truth hidden vector $\boldsymbol{z}^{vl}$ with error.

Formally, we say teacher has $\epsilon_v^n$ error if for all possible $n$ budget subsets $S \subset D_{\text{train}}$,

$$\frac{1}{|S|} \left\| \sum_{\boldsymbol{x}^{vl} \in S} \bar{G}_v^\top \boldsymbol{x}^v (\boldsymbol{x}^v)^\top \bar{G}_v - \sum_{\boldsymbol{x}^{vl} \in S} \boldsymbol{z}^v (\boldsymbol{z}^v)^\top \right\|_* \leq \epsilon_v^n$$

where the same notation applies for the language modal. By the orthonormal assumption on the ground truth matrix $G_{vl}^*$, we see that $\bar{G}_v^\top$ is aiming to inverting the map. In addition, we say the teacher has $\epsilon_{v*l}^n$ cross modal error

$$\frac{1}{|S|} \| \sum_{\boldsymbol{x}^{vl} \in S} \bar{G}_v^\top \boldsymbol{x}^v (\boldsymbol{x}^l)^\top \bar{G}_l - \sum_{\boldsymbol{x}^{vl} \in S} \boldsymbol{z}^v (\boldsymbol{z}^l)^\top \|_* \leq \epsilon_{v*l}^n$$

When all $\epsilon_v^n, \epsilon_l^n, \epsilon_{v*l}^n \to 0$ as $n \to \infty$, then we say the teacher is strong for both modalities. But it might also be possible that only one modal, for example, visual is strong. That is $\epsilon_v^n \to 0, \epsilon_l^n, \epsilon_{v*l}^n \gg \epsilon_v^n$.

**Model and training.** According to Lemma 4.1 in [46], using the CLIP loss to optimize the linear model has approximately the same training dynamics as using the regularized linear loss. Therefore, here we assume that we are learning $G_v, G_l$ by maximizing the clip score gap between the contrastive pairs, plus a regularizer,

$$\min_{G_v, G_l} \mathcal{L}_S^\rho(G_v, G_l) := \min_{G_v, G_l} \frac{\sum_{i \in S} \sum_{j \in S} (s_{ij} - s_{ii})}{|S|(|S| - 1)} + \frac{\rho}{2} \frac{|S|}{|S| - 1} \|G_v G_l^\top\|_F^2$$

where $s_{ij} := \langle G_v^\top \boldsymbol{x}_i^v, G_l^\top \boldsymbol{x}_j^l \rangle$ and $\rho > 0$ is some regularizer-related *constant*. Note that this objective maximizes self-similarity and minimizes similarity between disparate pairs. Note that this "loss" can be negative, avoiding the trivial null solution of all zeros. We denote this training process from any given $S$ as $G_{vl} = \mathcal{A}^\rho(S)$.

**Goal and metric.** Under the same principle as our training loss function, we measure the performance of any learnt $G_v, G_l$ on some downstream task with distribution $\mathcal{D}_{\text{target}}$ as test loss $\mathcal{L}_{\text{target}}(G_v, G_l) :=$

$$\mathbb{E}_{\substack{\boldsymbol{x}^{vl} \sim \mathcal{D}_{\text{target}} \\ \boldsymbol{x}_2^{vl} \sim \mathcal{D}_{\text{target}}}} (\langle G_v^\top \boldsymbol{x}^v, G_l^\top \boldsymbol{x}_2^l \rangle - \langle G_v^\top \boldsymbol{x}^v, G_l^\top \boldsymbol{x}^l \rangle)$$

This is inspired by the following classification accuracy. Assume that the test data including $C$ class, and the class distribution is $\mathcal{C}$. For every class $c$, the training data $\boldsymbol{x} = (\boldsymbol{x}^v, \boldsymbol{x}^l)$ satisfies distribution $\mathcal{P}_c$. We further assume the corresponding classification templates are $\{\boldsymbol{x}_c\}_{c=1}^C$. Thus we define classification accuracy as

$$\mathrm{AC}(G_v, G_l) = \mathbb{E}_{c,c' \sim \mathcal{C} \times \mathcal{C}} \left[ \mathbb{E}_{\boldsymbol{x}_i \sim \mathcal{P}_c} \mathbf{1}[s_{ic} > s_{ic'}] \right]$$

Therefore our goal is to minimize its gap between the best hind-side subset, for any $\rho$, without budget constraints,

$$\Delta^\rho(S) = \mathcal{L}_{\text{target}}(\hat{G}_{vl}) - \min_{S' \in D_{\text{train}}} \mathcal{L}_{\text{target}}(\mathcal{A}^\rho(S')), \hat{G}_{vl} = \mathcal{A}^\rho(S)$$

### A.2.2 Generalization Guarantees

We now provide theoretical guarantees and postpone our proof into Appendix A.2.3. **Firstly, we are going to prove the intuition behind NormSim$_2$score.**

**Lemma A.1** (Intuition behind NormSim$_2$). *With high probability at least $1 - \frac{1}{|S|^d}$, suppose the hind-side best subset has at least $\underline{n}$ number of samples, then we have*

$$\Delta^\rho(S) = \underbrace{\frac{1}{\rho} \max_{S' \in D_{train}} \left( \mathrm{Tr}\left( \Sigma_{target}(\Sigma_{S'} - \Sigma_S) \right) \right)}_{NormSim_2\,related\,term} + \underbrace{\mathcal{O}\left( \sqrt{\frac{d \log(d|S|)}{\underline{n}}} + \sqrt{\frac{d \log(d|S|)}{|S|}} \right)}_{noise}$$

*Proof sketch.* ❶ Under the assumption that both $\boldsymbol{z}^{vl}, \xi_{vl}$ is zero-mean, maximizing the clip score gap is equivalent to maximizing the clip score of the same sample.

$$\mathcal{L}_{\text{target}}(\hat{G}_v, \hat{G}_l) := -\mathbb{E}_{\boldsymbol{x}^{vl} \sim \mathcal{D}_{\text{target}}} \langle \hat{G}_v^\top \boldsymbol{x}^v, \hat{G}_l^\top \boldsymbol{x}^l \rangle$$

❷ By minimizing the regularized training loss $\mathcal{L}_S^\rho(G_v, G_l)$ using Eckart-Young-Mirsky Theorem, we get a closed form solution of $\hat{G}$ as

$$\hat{G}_v \hat{G}_l^\top \approx \frac{1}{\rho} G_v^* \Sigma_S \cdot (G_l^*)^\top + \text{noise depend on } S$$

❸ Combining the result in ❷ and ❶, we have

$$\mathcal{L}_{\text{target}}(\hat{G}_{vl}) \approx -\frac{1}{\rho} \mathrm{Tr}\left( \Sigma_{\text{target}} \Sigma_S \right) - \text{noise depend on } S$$

The same analysis can be applied on $\min_{S' \in D_{\text{train}}} \mathcal{L}_{\text{target}}(\mathcal{A}(S'))$ as well. Rearranging these two equations gives us the final result. □

This lemma shows the the $\Delta(S)$ is depend on the NormSim$_2$-related term and the noise term which comes from $\xi$. When $\underline{n}$ and $|S|$ is large enough, then the NormSim$_2$-related term will become dominant. This aligns with our practice experience that the final performance is less sensitive to the small variation in the number of select data as long as that is sufficient. Moreover, in some special cases where test distribution has identity cross-variance, then sampling by choosing CLIP score might be enough.

**Now we are ready to give a proof on the choice of $\bar{\Sigma}_{\text{target}}$ and visual-only information.** Specifically, the strategy error mainly comes from (1). The unknown test distribution shift from training. (2). The unobservable ground truth $\Sigma_S$. To tackle error (1), we assume some prior knowledge on test by using the proxy test variance $\bar{\Sigma}_{\text{target}}$. To tackle the error (2), there are two possible solutions as shown below. Based on the theoretical interpretation, we should choose different strategy based on the property of the teacher embedding model.

$$S_{\text{vision+language}} = \arg\max_S \mathrm{Tr}\left( \bar{\Sigma}_{\text{target}}(\sum_{\boldsymbol{x}^{vl} \in S} \bar{G}_v^\top \boldsymbol{x}^v (\boldsymbol{x}^l)^\top \bar{G}_l) \right)$$

$$S_{\text{vision only}} = \arg\max_S \mathrm{Tr}\left( \bar{\Sigma}_{\text{target}}(\sum_{\boldsymbol{x}^{vl} \in S} \bar{G}_v^\top \boldsymbol{x}^v (\boldsymbol{x}^v)^\top \bar{G}_v) \right)$$

**Theorem A.2** (Main). *Under the assumption of Lemma A.1,*

$$\Delta^\rho(S) \le noise + \frac{1}{\rho}\|\bar{\Sigma}_{target} - \Sigma_{target}\|\|\Sigma_S - \Sigma_{best}\|_*$$

$$+ \frac{1}{\rho}\begin{cases} \epsilon_{v*l}^S & \text{(vision+language)} \\ \epsilon_v^S + \sqrt{1 - \frac{1}{|S|}\sum_{i\in[S]}\langle z^v, z^l\rangle} & \text{(vision only)} \end{cases}$$

Firstly, it is evident that the greater the difference between $\bar{\Sigma}_{\text{target}}$ and $\Sigma_{\text{target}}$, the less improvement we can expect. Moreover, in scenarios where $\epsilon_l$ is large (indicating lower accuracy in the language part) while $\epsilon_v$ is small (indicating higher accuracy in the vision part), it may be advisable to opt for vision-only embeddings. However, the learner should also consider the term $\sqrt{1 - \frac{1}{|S|}\sum_{i\in S}\langle z^v, z^l\rangle}$, which represents the alignment between the ground truth visual and language latent vectors, essentially reflecting the intrinsic quality of the data. If this term is already significant, relying solely on vision information as a proxy for language information could lead to suboptimal results.

### A.2.3 Detailed proofs

**Lemma A.2.** *Let*

$$\hat{G}_v, \hat{G}_l = \arg\min_{G_v, G_l \in \mathbb{R}^{d\times r}} \mathcal{L}(G_v, G_l) \tag{7}$$

*Then we have*

$$\hat{G}_v\hat{G}_l^\top = \frac{1}{\rho}G_v^*\Sigma_S(G_l^*)^\top + P_1 + P_2 + P_3 + P_4 \tag{8}$$

*where noise terms $P_i$ are defined in (12) , (13), (14) and (15).*

*Proof.* Note that $s_{ij} = (\boldsymbol{x}_j^l)^\top G_l G_v^\top \boldsymbol{x}_i^v = \text{Tr}(G_v^\top \boldsymbol{x}_i^v(\boldsymbol{x}_j^l)^\top G_l)$, like the proof of Corollary B.1. in [46], we have

$$
\begin{aligned}
\mathcal{L}(G_v, G_l) &= \frac{\sum_{i\in S}\sum_{j\in S}(s_{ij} - s_{ii})}{|S|(|S|-1)} + \frac{\rho}{2}\frac{|S|}{|S|-1}\|G_v G_l^\top\|_F^2 \\
&= \frac{\sum_{i\in S}\sum_{j\in S} s_{ij} - |S|\sum_{i\in S} s_{ii}}{|S|(|S|-1)} + \frac{\rho}{2}\frac{|S|}{|S|-1}\|G_v G_l^\top\|_F^2 \\
&= -\text{Tr}\left(G_v^\top\left[\frac{1}{|S|-1}\sum_{i\in S}\boldsymbol{x}_i^v(\boldsymbol{x}_i^l)^\top - \frac{|S|}{|S|-1}\bar{\boldsymbol{x}}^v(\bar{\boldsymbol{x}}^l)^\top\right]G_l\right) + \frac{\rho}{2}\frac{|S|}{|S|-1}\|G_v G_l^\top\|_F^2 \\
&=: -\text{Tr}(G_v^\top\Gamma G_l) + \frac{\rho}{2}\frac{|S|}{|S|-1}\|G_v G_l^\top\|_F^2
\end{aligned}
$$

where $\bar{\boldsymbol{x}}^{vl} := (\sum_{i\in S}\boldsymbol{x}_i^{vl})/|S|$. Then by the Eckart-Young-Mirsky Theorem (For example, Theorem 2.4.8 in Golub et al. [47]), we know that

$$
\begin{aligned}
&\arg\min_{G_v\in\mathbb{R}^{d\times r}, G_l\in\mathbb{R}^{d\times r}} \mathcal{L}(G_v, G_l) \\
&= \arg\max_{G_v\in\mathbb{R}^{d\times r}, G_l\in\mathbb{R}^{d\times r}} \text{Tr}(G_v^\top\Gamma G_l) - \frac{\rho}{2}\frac{|S|}{|S|-1}\|G_v G_l^\top\|_F^2 \\
&= \left\{(G_v, G_l)\in\mathbb{R}^{d\times r}\times\mathbb{R}^{d\times r} : G_v G_l^\top = \frac{1}{\rho}\frac{|S|-1}{|S|}\text{SVD}_r(\Gamma)\right\} \quad \text{(Eckart-Young-Mirsky Theorem)}
\end{aligned}
$$

where the notation $\text{SVD}_r(\Gamma)$ means choosing the first $r$ components of the matrix $\Gamma$. Further note that

$$
\begin{aligned}
\Gamma &= \frac{1}{|S|-1}\sum_{i\in S}\boldsymbol{x}_i^v(\boldsymbol{x}_i^l)^\top - \frac{|S|}{|S|-1}\bar{\boldsymbol{x}}^v(\bar{\boldsymbol{x}}^l)^\top \tag{9} \\
&=: P_0 + P_1 + P_2 + P_3 + P_4 \tag{10}
\end{aligned}
$$

Here note that $\Sigma_S = \frac{1}{|S|}\sum_{i\in S}\boldsymbol{z}_i^v(\boldsymbol{z}_i^l)^\top$, we have $P_i$ as follows:

$$P_0 \quad:=\quad \frac{|S|}{|S|-1}G_v^*\cdot\Sigma_S\cdot(G_l^*)^\top \tag{11}$$

$$P_1 \quad:=\quad \frac{1}{|S|-1}G_v^*\sum_{i\in S}\boldsymbol{z}_i^v(\boldsymbol{\xi}_i^l)^\top \tag{12}$$

$$P_2 \quad:=\quad \frac{1}{|S|-1}\sum_{i\in S}\boldsymbol{\xi}_i^v(\boldsymbol{z}_i^l)^\top(G_l^*)^\top \tag{13}$$

$$P_3 \quad:=\quad \frac{1}{|S|-1}\sum_{i\in S}\boldsymbol{\xi}_i^{(1)}(\boldsymbol{\xi}_i^{(2)})^\top \tag{14}$$

$$P_4 \quad:=\quad -\frac{|S|}{|S|-1}\bar{\boldsymbol{x}}^v(\bar{\boldsymbol{x}}^l)^\top \tag{15}$$

It's clear that the rank of the matrix $P_0$ is no more than $r$, so $\mathrm{SVD}_r(P_0) = P_0$. And for $i\in\{1,2,3,4\}$, $P_i$ are noise terms with $\mathbb{E}[P_i] = O$. $\qquad\square$

**Lemma A.3.** *For any fixed $S$, w.h.p $1-\delta$ the noise term can be upper bounded by $\sqrt{\frac{d\log(1/\delta)}{|S|}}$*

*Proof.* To upper bound the P1 and P2, we have

$$\|\sum_i \boldsymbol{z}_i^{vl}(\xi_i^{vl})^\top\|_*^2 = \mathrm{Tr}\left(\sum_{i,j}\xi_i^{vl}(\boldsymbol{z}_i^{vl})^\top\boldsymbol{z}_j^{vl}\xi_j^{vl}\right) = \sum_{i,j}(\boldsymbol{z}_i^{vl})^\top\boldsymbol{z}_j^{vl}(\xi_j^{vl})^\top\xi_i^{vl}$$

$$\mathbb{E}\|\sum_i \boldsymbol{z}_i^{vl}(\xi_i^{vl})^\top\|_*^2 = \mathbb{E}\left[\sum_i(\boldsymbol{z}_i^{vl})^\top\boldsymbol{z}_i^{vl}(\xi_i^{vl})^\top\xi_i^{vl}\right] = |S|d$$

Regarding each $(\boldsymbol{z}_i^{vl})^\top\boldsymbol{z}_j^{vl}(\xi_j^{vl})^\top\xi_i^{vl}$ as weakly dependent variable, then by using Bernstein inequality, we have, with high probability $1-\delta$,

$$\|\sum_i \boldsymbol{z}_i^{vl}(\xi_i^{vl})^\top\|_*^2 \le |S|d + \sqrt{d|S|^2\sigma_\xi^2\log(1/\delta)} \le |S|d\sqrt{\log(1/\delta)}$$

So $\frac{1}{|S|}\|\sum_i \boldsymbol{z}_i^{vl}(\xi_i^{vl})^\top\|_* \le \sqrt{\frac{d\log(1/\delta)}{|S|}}$. Note that $\|\bar{\boldsymbol{x}}^{vl}\| \lesssim \sqrt{\frac{\log(|S|d)}{|S|}}$ (like Proposition 2.5 in Wainwright et al. [48]), it is easy to see that P3 ad P4 are the low order terms if $\delta \lesssim \frac{1}{|S|d}$. $\qquad\square$

**Lemma A.4** (Intuition behind VAS). *With high probability $1-\delta$, suppose the hind-side best subset has at least $\underline{n}$ number of samples, then we have*

$$\Delta(S) = \frac{1}{\rho}\max_{S'\in D_{train}}\left(\mathrm{Tr}\left(\Sigma_{target}(\Sigma_{S'}-\Sigma_S)\right)\right) + \sqrt{\frac{d\log(1/\delta)}{\underline{n}}} + \sqrt{\frac{d\log(1/\delta)}{|S|}}$$

*Proof.* For any learnt $G_v, G_l$ based on dataset $S$, we have

$$\begin{aligned}
\mathcal{L}_{\text{test}}(G_v, G_l) &= \mathrm{Tr}(G_v^\top\mathbb{E}_{\boldsymbol{x}_{vl}\sim\mathcal{D}_{\text{target}}}[\boldsymbol{x}^v(\boldsymbol{x}^l)^\top]G_l)\\
&= \mathrm{Tr}(\mathbb{E}_{\boldsymbol{x}_{vl}\sim\mathcal{D}_{\text{target}}}[\boldsymbol{x}^v(\boldsymbol{x}^l)^\top]G_lG_v^\top)\\
&= \frac{1}{\rho}\mathrm{Tr}\left(\mathbb{E}_{\boldsymbol{x}_{vl}\sim\mathcal{D}_{\text{target}}}[\boldsymbol{x}^v(\boldsymbol{x}^l)^\top]G_l^*\Sigma_S(G_v^*)^\top\right) - \mathrm{Tr}\left(\mathbb{E}_{\boldsymbol{x}_{vl}\sim\mathcal{D}_{\text{target}}}[\boldsymbol{x}^v(\boldsymbol{x}^l)^\top]\text{noise}_S\right)\\
&= \frac{1}{\rho}\mathrm{Tr}\left((G_v^*)^\top\mathbb{E}_{\boldsymbol{x}_{vl}\sim\mathcal{D}_{\text{target}}}[\boldsymbol{x}^v(\boldsymbol{x}^l)^\top]G_l^*\Sigma_S\right) - \mathrm{Tr}\left(\mathbb{E}_{\boldsymbol{x}_{vl}\sim\mathcal{D}_{\text{target}}}[\boldsymbol{x}^v(\boldsymbol{x}^l)^\top]\text{noise}_S\right)\\
&= -\frac{1}{\rho}\mathrm{Tr}\left(\Sigma_{\text{target}}\Sigma_S\right) - \mathrm{Tr}\left(\mathbb{E}_{\boldsymbol{x}_{vl}\sim\mathcal{D}_{\text{target}}}[\boldsymbol{x}^v(\boldsymbol{x}^l)^\top]\text{noise}_S\right)
\end{aligned}$$

Here the first equation comes from Theorem A.4 and the third equation comes from Lemma A.2. Consequently, we have

$$-\min_{S'\in D_{\text{train}}} \mathcal{L}_{\text{test}}(\mathcal{A}(S')) = \max_{S'\in D_{\text{train}}} \left(\frac{1}{\rho}\operatorname{Tr}\left(\Sigma_{\text{target}}\Sigma_{S'}\right) + \operatorname{Tr}\left(\mathbb{E}_{\boldsymbol{x}_{vl}\sim\mathcal{D}_{\text{target}}}[\boldsymbol{x}^v(\boldsymbol{x}^l)^\top]\text{noise}_{S'}\right)\right)$$

$$\leq \frac{1}{\rho}\max_{S'\in D_{\text{train}}}\left(\operatorname{Tr}\left(\Sigma_{\text{target}}\Sigma_{S'}\right)\right) + \|\mathbb{E}_{\boldsymbol{x}_{vl}\sim\mathcal{D}_{\text{target}}}[\boldsymbol{x}^v(\boldsymbol{x}^l)^\top]\|\|\text{noise}_{S'}\|_*$$

$$\leq \frac{1}{\rho}\max_{S'\in D_{\text{train}}}\left(\operatorname{Tr}\left(\Sigma_{\text{target}}\Sigma_{S'}\right)\right) + \mathcal{O}\left(\sqrt{\frac{d\log(1/\delta)}{n}}\right)$$

Therefore, we have the final result as

$$\Delta(S) = \mathcal{L}_{\text{test}}(\hat{G}_{vl}) - \min_{S'\in D_{\text{train}}}\mathcal{L}_{\text{test}}(\mathcal{A}(S'))$$

$$= \frac{1}{\rho}\max_{S'\in D_{\text{train}}}\left(\operatorname{Tr}\left(\Sigma_{\text{target}}(\Sigma_{S'} - \Sigma_S)\right)\right) + \mathcal{O}\left(\sqrt{\frac{d\log(1/\delta)}{n}} + \sqrt{\frac{d\log(1/\delta)}{|S|}}\right)$$

$$\square$$

**Theorem A.3** (Main). *Under the assumption of Lemma A.1, we have*

$$\Delta(S) \leq noise + \|\bar{\Sigma}_{target} - \Sigma_{target}\|\|\Sigma_S - \Sigma_{best}\|_*$$

$$+ \begin{cases} \epsilon_{v*l}^S & (vision+language) \\ \left(\epsilon_v^S + \sqrt{1 - \frac{1}{|S|}\sum_{i\in[S]}\langle\boldsymbol{z}^v,\boldsymbol{z}^l\rangle}\right) & (vision only) \end{cases}$$

*Proof.* Based on Lemma A.1, we will focus on the error cause from selecting subset $S$, that is, $\operatorname{Tr}\Sigma_{\text{target}}\Sigma_S$. Since the exact $\Sigma_{\text{target}}$ is unknown, we assume the access to some proxy $\bar{\Sigma}_{\text{target}}$ instead.

Recall that, for any $S$, we have ground-truth $\Sigma_S = \mathbb{E}_{\boldsymbol{z}_{vl}\in S}\boldsymbol{z}^v(\boldsymbol{z}^l)^\top$. Unfortunately, this is not directly observable by the learner. Instead, the learner is able to observe some proxy $\bar{\Sigma}_S$ based on the teacher model $\bar{G}_{vl}$ and therefore solving

$$\arg\max_S \operatorname{Tr}\left(\bar{\Sigma}_{\text{target}}\bar{\Sigma}_S\right)$$

and therefore, denote $\Sigma_{\text{best}} = \arg\max_{S'\in D_{\text{train}}} \operatorname{Tr}\left(\Sigma_{\text{target}}\Sigma_{S'}\right)$

$$\operatorname{Tr}\left(\Sigma_{\text{target}}(\Sigma_{\text{best}} - \Sigma_S)\right) = \operatorname{Tr}\left(\bar{\Sigma}_{\text{target}}(\Sigma_{\text{best}} - \bar{\Sigma}_S)\right) + \operatorname{Tr}\left(\bar{\Sigma}_{\text{target}}(\bar{\Sigma}_S - \Sigma_S)\right) + \operatorname{Tr}\left((\Sigma_{\text{target}} - \bar{\Sigma}_{\text{target}})(\Sigma_{\text{best}} - \Sigma_S)\right)$$

$$\leq \operatorname{Tr}\left(\bar{\Sigma}_{\text{target}}(\bar{\Sigma}_S - \Sigma_S)\right) + \operatorname{Tr}\left((\Sigma_{\text{target}} - \bar{\Sigma}_{\text{target}})(\Sigma_{\text{best}} - \Sigma_S)\right)$$

$$\leq \|\bar{\Sigma}_{\text{target}}\|\|\bar{\Sigma}_S - \Sigma_S\|_* + \|\bar{\Sigma}_{\text{target}} - \Sigma_{\text{target}}\|\|\Sigma_S - \Sigma_{\text{best}}\|_*$$

where the first inequality is by the definition of $\bar{\Sigma}_S$ and the second inequality comes from holder's inequality. Now the key is to upper bound $\|\bar{\Sigma}_S - \Sigma_S\|_*$ based on our chosen strategy.

In option 1, we use the clip embedding from both visual and language modal. That is, choose $\bar{\Sigma}_S = \sum_{\boldsymbol{x}_{vl}\in S}(\bar{G}_v)^\top\boldsymbol{x}^v(\boldsymbol{x}^l)^\top\bar{G}_l$. Then we have

$$\|\bar{\Sigma}_S - \Sigma_S\|_* \leq \frac{1}{|S|}\|\sum_{\boldsymbol{x}_{vl}\in S}(\bar{G}_v)^\top\boldsymbol{x}^v(\boldsymbol{x}^l)^\top\bar{G}_l - \sum_{\boldsymbol{x}_{vl}\in S}\boldsymbol{z}^v(\boldsymbol{z}^l)^\top\|_* \leq \epsilon_{v*l}^S$$

In option 2, we use the clip embedding from language model only. That is choose $\bar{\Sigma}_S = \sum_{\boldsymbol{x}_{vl}\in S}\bar{G}_v^\top\boldsymbol{x}^v(\boldsymbol{x}^v)^\top\bar{G}_v$. Then, by definition of $\epsilon_S$, we have

$$\|\bar{\Sigma}_S - \Sigma_S\|_* \leq \frac{1}{|S|}\|\sum_{\boldsymbol{x}_{vl}\in S}\bar{G}_v^\top\boldsymbol{x}^v(\boldsymbol{x}^v)^\top\bar{G}_v - \sum_{\boldsymbol{x}_{vl}\in S}\boldsymbol{z}^v(\boldsymbol{z}^v)^\top\|_* + \frac{1}{|S|}\|\sum_{\boldsymbol{x}_{vl}\in S}\boldsymbol{z}^v(\boldsymbol{z}^v)^\top - \Sigma_S\|_*$$

$$\leq \epsilon_v^S + \frac{1}{|S|}\|\sum_{\boldsymbol{x}_{vl}\in S}\boldsymbol{z}^v(\boldsymbol{z}^v)^\top - \Sigma_S\|_*$$

Now to further bound the second term, we have

$$\frac{1}{|S|}\|\sum_{\boldsymbol{x}_{vl}\in S}\boldsymbol{z}^v(\boldsymbol{z}^v)^\top - \Sigma_S\|_* \le \frac{1}{|S|}\|Z_v^\top\|_*\|Z_v - Z_l\|_*$$

$$= \frac{1}{|S|}\sqrt{\operatorname{Tr} Z_v Z_v^\top}\sqrt{\operatorname{Tr}(Z_v - Z_l)^\top(Z_v - Z_l)}$$

$$= \frac{1}{|S|}\sqrt{\operatorname{Tr}(I_{n\times n})}\sqrt{2\operatorname{Tr}\left(I_{n\times n} - Z_v Z_l^\top\right)}$$

$$= \frac{1}{|S|}\sqrt{2|S|(|S| - \sum_{i\in[S]}\langle\boldsymbol{z}^v,\boldsymbol{z}^l\rangle)}$$

$$= \sqrt{1 - \frac{1}{|S|}\sum_{i\in[S]}\langle\boldsymbol{z}^v,\boldsymbol{z}^l\rangle)}$$

Therefore, we finish the proof. $\qquad\square$

**Theorem A.4** (A simplified version of test loss)**.** *Under the assumption that both $\boldsymbol{z}_{vl}, \xi_{vl}$ is zero-mean, maximizing the clip score gap is equivalent to maximize the clip score of the same sample.*

$$\mathcal{L}_{target}(G_v, G_l) := -\mathbb{E}_{\boldsymbol{x}_{vl}\sim\mathcal{D}_{target}}\langle G_v^\top\boldsymbol{x}_v, G_l^\top\boldsymbol{x}_l\rangle$$

*Proof.* For any $\boldsymbol{x}_{vl}$, we have

$$\mathbb{E}_{\boldsymbol{x}'_{vl}\sim\mathcal{D}_{target}}(\langle G_v^\top\boldsymbol{x}_v, G_l^\top\boldsymbol{x}'_l\rangle - \langle G_v^\top\boldsymbol{x}_v, G_l^\top\boldsymbol{x}_l\rangle)$$

$$= \langle G_v^\top\boldsymbol{x}_v, G_l^\top\mathbb{E}_{\boldsymbol{x}'_{vl}\sim\mathcal{D}_{target}}(\boldsymbol{x}'_l - \boldsymbol{x}_l)\rangle$$

$$= -\langle G_v^\top\boldsymbol{x}_v, G_l^\top\boldsymbol{x}_l\rangle$$

$\qquad\square$

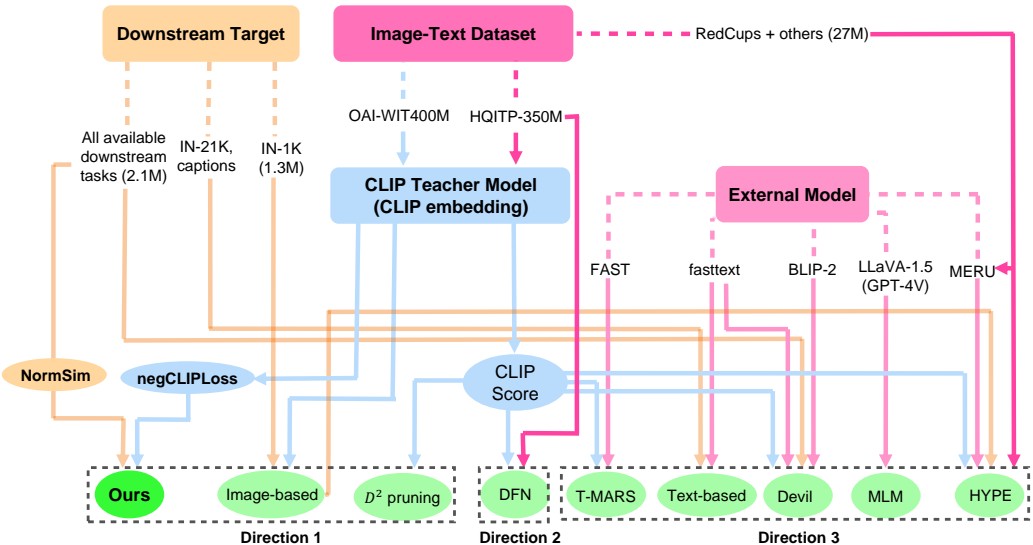

Figure 4: Illustration of different directions for data selection methods for multimodal contrastive learning. Here we use four colors to denote the four main resources we can obtain: CLIP teacher model, downstream target data (which is much smaller than the external multimodal dataset or pretraining dataset), the external image-text dataset, and the external non-CLIP model. **Direction 1** denotes the methods that only use the original OAI CLIP teacher model and the downstream target data. **Direction 2** represents the methods that use external datasets to train a new CLIP teacher model for improving filtering, like DFN [2]. **Direction 3** denotes the methods that use external non-CLIP model to select the data that may be heuristically helpful for downstream tasks, like image without too much text or be more special. In general, *D1 method using only CLIP embedding, like negCLIPLoss, is orthogonal to D2. And both D1 and D2 can be combined with D3 to explore better filtering results.* In the experiments part of the main paper (Sec. 4), we further show that our proposed D1 methods: NormSim and negCLIPLoss, can outperform all the D3 baselines except the best method "HYPE ∪ DFN". And we can achieve the new state-of-the-art by combining our methods with that method.

## B    Illstration of Different Directions for Data Selection in Multimodal Contrastive Learning

We summarize our main idea of categorizing the current top data selection methods in Figure 4.

## C    Details of Experiments

### C.1    Computation Cost

Our algorithm can significantly reduce the computational cost compared to many existing works as shown in Table 5. For example, when the CLIP embeddings are obtained (cost about 50 hours for CLIP-B/32), both T-MARS [12] and MLM [42] still require more than 900 hours data pre-processing time to extract the required information from 110M size dataset of DataComp-medium, while we only need about 5 hours. On the other hand, DFN, although has a similar forward speed (i.e. preprocessing time), requires retraining a new CLIP teacher model on the HQITP-350M, which is larger than DataComp-medium.

We give some details in estimating the preprocessing time of other methods:

- For **T-MARS** and $\mathbb{D}^2$ pruning, we run their official code on DataComp-small (11M) data, and simply scale the preprocessing time by 10 for DataComp-medium, given that the preprocessing time for T-MARS is proportional to the size of the pretraining dataset, while $\mathbb{D}^2$ pruning is no faster than linear.

Table 5: Comparison of preprocessing time and external resources needed between our method and other D3 category methods. We skip DFN since it's orthogonal to our negCLIPLoss method and we can directly improve it as mentioned in Table 1. Here since all the baselines below except MLM use a pretrained CLIP model, we only count the time that doesn't contain that for inferring CLIP image/text embeddings (about 50 L40 hours for OAI CLIP-B/32), which is also adopted in DataComp benchmark [1]. The external dataset corresponds to the external multimodal dataset used for training or finetuning the external model. Notably, the preprocessing time for the following methods are all approximately linearly proportional to the amount of unfiltered pretrained dataset.

| Type | Filtering Strategy | Ext. Model Used | Size of Ext. Dataset | Preprocess Time | Training Time | Avg. |
|------|--------------------|------------------|----------------------|-----------------|---------------|------|
| D1 | $\mathbb{D}^2$ Pruning [18] | NA | NA | >70 L40 h | 65 L40 h | 29.5 |
| D3 | T-MARS [12] | FAST [13] | NA | 950 L40 h | 65 L40 h | 34.1 |
| D3 | MLM [42] | LLaVA-1.5 [43, 44] | 50k | 1120 A100 h | 65 L40 h | 34.5 |
| D3 | Devil [14] | fasttext [15], BLIP-2 [16] | NA | 510 A100 h | 65 L40 h | 34.5 |
| D3 | HYPE [3] | MERU [45] | 27M | > 120 L40 h | 65 L40 h | 31.9 |
| D1 | **Ours (20%)** | NA | NA | **5 L40 h** | 65 L40 h | **35.2** |

- For **MLM**, we get the estimated time from their paper. They mention that they need 6.1 minutes to process 10k samples on A100, which results in 1120 A100 hours for our dataset (110M). We need to mention that their estimation time of calculating CLIP embedding is inaccurate and we can do it much faster than their claim using the DataComp pipeline.

- For **Devil**, it needs to run the k-means clustering algorithm from the faiss library on the embedding space, which is estimated to cost 120 L40 hours on DataComp-medium. Using BLIP-2 [16] to scan the whole dataset will need about 470 A100 hours from the experimental details in [17]. From https://lambdalabs.com/gpu-benchmarks, we roughly assume that 120 L40 hours are at least comparable to 40 A100 hours for K-means clustering.

- For **HYPE**, they claim that MERU is as efficient as CLIP, but they still need at least 120 L40 hours for processing 110M data for their final score, since it uses the image embedding clusters on DataComp-medium obtained from running k-means clustering algorithm.

## C.2  Details of negCLIPLoss

We give the pseudocode of calculating negCLIPLoss in Algorithm 1, which is specially designed for pytorch-style parallel matrix calculation. It can be fully accelerated and the computation cost introduced by the normalization term is negligible compared with the training time or preprocessing time of other top baselines as detailed in Table C.1.

In negCLIPLoss, we need to get the batch size $|B|$ and the value of the learnable temperature parameter $\tau$ at the final step of the teacher model pretraining stage. For OAI CLIP-L/14 and OAI CLIP-B/32, these values are $\tau = 0.01$ and $|B| = 32768$.

We also have an ablation study about the temperature parameter and batch size chosen for CLIP teacher models as shown in Table 6. We will see that in general, a larger batch size will result in better performance, and $\tau = 0.01, b = 32768$ is the best choice for both OAI CLIP-B/32 and DFN-P. The reason for such a batch size is that a larger batch can contain more contrastive data pairs, which is also supported by the concentration result of the normalization term proved in Appendix A.1, and thus it can check the image-text matching between more different data. Therefore, we always consider the largest batch size 32768 which can fit into a single 24G GPU in the CLIP forward pass, which is also the OAI CLIP training batch size.

## C.3  Details of NormSim$_2$-D

In this section, we illustrate the details of our NormSim$_2$-D algorithm. The top-$N$ selection method is aiming to achieve the object:

$$S = \arg \max_{|S|=N} \sum_{i \in S} \bar{f}_v(x_i^v)^\top \left( \frac{1}{|X_{\text{target}}|} \sum_{x_t \in X_{\text{target}}} \bar{f}_v(x_t^v) \bar{f}_v(x_t^v)^\top \right) \bar{f}_v(x_i^v) \tag{16}$$

Table 6: Ablation study about the temperature parameters $\tau$ and batch size $b$ for CLIP teacher model. The values obtained from the last training step of the teacher models are $\tau = 0.01, b = 32768$ for OAI CLIP-B/32, OAI CLIP-L/14, and $b = 16384, \tau = 0.07$ for DFN-P. In the main paper, we use $b = 32768, \tau = 0.01$ for all three kinds of teacher models.

| OAI CLIP-B/32 | Size | IN-1k | IN Dist. Shift | VTAB | Retr. | Avg. |
|---|---|---|---|---|---|---|
| **CLIPScore (30%)** [38] | 33M | 27.6 | 24.2 | 33.6 | 25.1 | 33.2 |
| **negCLIPLoss (30%)** | | | | | | |
| $b = 16384, \tau = 0.01$ | 33M | **28.8** | 25.0 | 32.5 | 26.2 | 33.0 |
| $b = 16384, \tau = 0.02$ | 33M | 28.6 | 24.8 | 33.3 | 25.3 | 33.1 |
| $b = 16384, \tau = 0.07$ | 33M | 28.0 | 24.2 | 33.5 | 25.1 | 32.6 |
| $b = 32768, \tau = 0.001$ | 33M | 16.0 | 13.9 | 25.1 | 19.4 | 24.4 |
| $b = 32768, \tau = 0.005$ | 33M | 28.5 | 25.0 | 33.6 | **27.0** | 33.0 |
| $b = 32768, \tau = 0.01$ | 33M | **28.8** | **25.1** | 33.7 | 26.6 | **33.6** |
| $b = 32768, \tau = 0.02$ | 33M | 28.5 | 24.8 | 33.6 | 26.2 | 32.9 |
| $b = 32768, \tau = 0.07$ | 33M | 28.2 | 24.5 | 32.8 | 25.2 | 32.7 |
| **negCLIPLoss (30%) $\cap$ NormSim$_\infty$(Target)** | | | | | | |
| $b = 16384, \tau = 0.01$ | 22M | **32.4** | **27.4** | 34.5 | 26.1 | 34.7 |
| $b = 16384, \tau = 0.02$ | 22M | 31.8 | 26.7 | 35.0 | 24.9 | 34.2 |
| $b = 16384, \tau = 0.07$ | 22M | 31.0 | 26.3 | 35.0 | 25.5 | 33.9 |
| $b = 32768, \tau = 0.005$ | 22M | 32.2 | 27.2 | 35.3 | **26.5** | 34.8 |
| $b = 32768, \tau = 0.01$ | 22M | **32.4** | **27.4** | 35.9 | 26.3 | **35.2** |

| DFN-P | Size | IN-1k | IN Dist. Shift | VTAB | Retr. | Avg. |
|---|---|---|---|---|---|---|
| **negCLIPLoss** | | | | | | |
| $15\%, b = 16384, \tau = 0.07$ | 16M | 31.0 | 27.0 | 35.2 | 26.8 | 34.2 |
| $15\%, b = 32768, \tau = 0.01$ | 16M | **31.3** | 27.3 | **35.8** | 26.4 | 34.6 |
| $17.5\%, b = 16384, \tau = 0.07$ | 19M | **31.3** | 27.2 | 33.5 | **27.6** | 33.5 |
| $17.5\%, b = 32768, \tau = 0.01$ | 19M | 31.2 | **27.5** | 35.7 | 27.0 | **34.7** |
| **negCLIPLoss (17.5%) $\cap$ NormSim$_\infty^{\text{B/32}}$(Target)** | | | | | | |
| $b = 16384, \tau = 0.07$ | 16M | 31.1 | **27.4** | 34.8 | **26.1** | 34.2 |
| $b = 32768, \tau = 0.01$ | 16M | **31.6** | 27.3 | **37.2** | 25.5 | **35.7** |

when the actual $X_{\text{target}}$ is unknown. In practice, removing one data at a time is too slow. Therefore, we remove a batch of data for every step. In detail, if the number of steps is $\tau$, and let $\bar{\Sigma}_{\text{test},i} = \frac{1}{|S_i|} \sum_{j \in S_i} \bar{f}_v(\boldsymbol{x}_j^v) \bar{f}_v(\boldsymbol{x}_j^v)^\top$ where $S_i$ is the selected subset at step $i$, then we will remove the data satisfies the following equation step-by-step until reaching the final subset size:

$$S_i \setminus S_{i+1} = \arg \min_{x_l \in S_i} \left[ \bar{f}_v(x_l^v)^T \cdot \left( \frac{1}{|S_i|} \sum_{x_t \in S_i} \bar{f}_v(x_t^v) \bar{f}_v(x_t^v)^\top \right) \cdot \bar{f}_v(x_l^v) \right], \quad i \in \{0, \ldots, \tau - 1\}$$

Then we can detail the algorithm process of NormSim$_2$-D in Algorithm 2. In general, the smaller the step size, the better the results. But in experiments, we find that it's already enough to get good results when $\tau = 500$.

## C.4 Details of Related Works

We add some details about the baselines used in our paper as follows.

- **Text-based filtering.** [1] proposes a text-based filtering that tries to select the data that contains caption overlapping with the class name from ImageNet-21K or ImageNet-1K.

- **Image-based filtering.** [1] also proposes a heuristic way to sample the visual content overlaps with ImageNet-1K classes. They first apply filtering by language (only choose English caption by fasttext [15]) and caption length (over two words and 5 characters). Then they cluster the image embeddings from training data to 100K groups using Faiss [49], and keep the groups whose cluster center is the nearest neighbor to at least one image embedding of ImageNet-1K image.

---

**Algorithm 1** negCLIPLoss

---

**Inputs:** image/text embeddings of the pretraining data $F^{vl} = [\{\bar{f}_{vl}(x_1^{vl})\}, \ldots, \{\bar{f}_{vl}(x_N^{vl})\}]^\top \in \mathbb{R}^{N \times d}$, batch size $b$, temperature parameter $\tau$, the number of times negCLIPLoss is random $K(= 10)$.

Initialize negCLIPLoss array $\boldsymbol{r} = [0, \ldots, 0] \in \mathbb{R}^N$

**for** $k = 1$ **to** $K$ **do**

    Get a random batch division $S_k = \{B_1, \ldots, B_s\}$ such that $s = \lceil N/b \rceil$. Every $B_i \in S_k$ is the index of a batch of data.

    **for** $j = 1$ **to** $s$ **do**

        Get batch of embeddings in batch $j$: $F_j^{vl} = F^{vl}[B_j] \in \mathbb{R}^{b \times d}$

        Get the similarity matrix: $E_j = F_j^v (F_j^l)^\top \in \mathbb{R}^{b \times b}$

        Get the CLIPScores: $\boldsymbol{c}_j = \mathrm{diag}(E_j) \in \mathbb{R}^b$

        Define $G_j = \exp(E_j/\tau)$

        Define $\boldsymbol{g}_j^v \in \mathbb{R}^b$ be the vector containing the sum of each row vector in $G_j$ (i.e., over image).

        Define $\boldsymbol{g}_j^l \in \mathbb{R}^b$ be the vector containing the sum of each column vector in $G_j$ (i.e., over text).

        Get the negCLIPLoss: $\boldsymbol{r}[B_j] = \boldsymbol{c}_j - 0.5\tau \cdot (\log(\boldsymbol{g}_j^v) + \log(\boldsymbol{g}_j^v))$, here we use element-wise operation.

    **end for**

**end for**

Take the mean of each random division as output: negCLIPLoss $= \boldsymbol{r}/K$

---

 

---

**Algorithm 2** NormSim-D strategy

---

**Inputs:** image embeddings of the data after CLIP score filtering $\{\bar{f}_v(x_i^v)\}_{i \in S}$, target size $N$, number of steps $\tau$

Initialize $S_0 = S, N_0 = |S|$

**for** $t = 1$ **to** $\tau$ **do**

    Size at step $t$ : $N_t = N_0 - \frac{t}{\tau}(N_0 - N)$.

    Prior matrix: $\bar{\Sigma}_{\text{test},t-1} = \sum_{j \in S_{t-1}} \bar{f}_v(x_j^v) \bar{f}_v(x_j^v)^\top$

    Updated NormSim$_2$-D for each sample $i$ in $S_{t-1}$:

$$\text{NormSim}_2\text{-D}(x_i) = \bar{f}_v(x_i^v)^\top \cdot \bar{\Sigma}_{\text{test},t-1} \cdot \bar{f}_v(x_i^v)$$

    Construct $S_t$ such that it contains the data with highest NormSim$_2$-D in $S_{t-1}$ and satisfies $|S_t| = N_t$.

**end for**

---

- $\mathbb{D}^2$ **Pruning.** [18] tries to represent the dataset as an undirected graph for coreset selection. They assign the difficulty for each example and use message passing to update the difficulty score incorporating the difficulty of its neighboring examples, and finally try to keep both diverse and difficult subsets. For our experiments, we adhere to the default hyperparameters of $\mathbb{D}^2$ on DataComp as specified in their official codebase.

- **T-MARS** [12] uses a text detection model like FAST [13] to filter out the data that only contain the texts of caption in the image and don't have other useful image features.

- **Devils** [14] combines many ways for data filtering. At the very first it filter data based on heuristic rules like text length, frequency of texts, and image size, and it also use CLIPScore for cross-modality matchment. Then it adopts target distribution alignment methods similar to image-based filtering, but instead of using ImageNet-1k only, it uses 22 downstream tasks as the target set. Further, it adopts external models fasttext [15] to remove non-English captions and image-captioning model BLIP-2 [50] to select images with MNIST-style digits.

- **MLM** [42] prompts GPT-4V to construct instruction data including the image-text data, and use it to fine-tune a smaller vision-language model like LLaVA-1.5 [43, 44] into a filtering network. Nevertheless, the number of parameters of LLaVA-1.5 is still much larger than CLIP, and thus LLaVA-1.5 has a much longer preprocessing time as mentioned in Table C.1.

## C.5 How to Choose Hyperparameters

The main hyper-parameters of our negCLIPLoss and NormSim are the target numbers for filtering (refer to Appendix C.2 for the setting of temperature and batch size), which is also the main concerns for all the top baselines like DFN, MLM, and T-MARS. In the case of DataComp settings, noting that all the top baselines in DataComp-medium benchmark keep the downsampling ratios ranging from 15% 30% to achieve the best results, we can set the sampling ratio as some previous baselines. Our method with OAI CLIP teacher model first selects the data with the top 30% negCLIPLoss, and then selects the top 66.7% NormSim scores to keep 20% of the original pool. We don't tune the target size carefully here for fair comparison.

In more general cases, we can recommend some **training-dataset-independent** thresholds for Norm-Sim, since the scores only depends on the norm $p$ and target data rather than other data in the pool. We recommend to set the threshold as 0.7 for $\text{NormSim}_\infty$(Target) and 0.15 for $\text{NormSim}_2$(IN-1k) in general. On the other hand for negCLIPLoss, note that like NormSim, CLIPScore is also training-dataset-independent, we recommend to first find the percentile of the data with CLIPScore=0.21, and then downsample the dataset using CLIPLoss until that particular percentile.

Overall, finding optimal filtering ratio for data selection algorithm is always difficult and out of the scope of this paper. From the paper about the scaling law for data filtering [51], downsampling size even depends on the computation budget. When you have more budget, you should sample more data for learning. And thus another possible solution is to use their fitting formula to get some recommended downsampling ratios.

At last, we also note that *in data selection problem, visualization is a simple but effective way for tuning parameters or finding downsampling ratios*. People can first randomly select a small subset (like 1000 data) on some pretraining data subset, and then calculate the target scores (CLIPScore, negCLIPLoss, NormSim or any other metrics) on them, and fianlly visualize the data corresponding to scores at different percentiles, like top 10%, 30%, 50% and 70% of the negCLIPLoss. In this way, we can determine the threshold of filtering directly by observing the data. We also give some visualization examples of our methods in Appendix E, We believe this is an effective way to give some guidance on how to roughly select the initial downsampling ratios.

## C.6 Discussion of NormSim

### C.6.1 How NormSim$_2$ Connects to Selecting the Data in Principal Components.

For convenience, we let $f(x_t)$ denote the image embedding of the target data $x_t \in X_T$, and $f(x_s)$ denotes the image embeddings of training data $x_s \in X_S$. Then the definition of NormSim on a data $x_s$ is

$$\text{NormSim}_p(X_T, x_s) = \left( \sum_{x_t \in X_T} [f(x_t)^\top f(x_s)]^p \right)^{1/p} \tag{17}$$

Then when $p = 2$, we have

$$\text{NormSim}_2(X_T, x_s) = \left( \sum_{x_t \in X_T} [f(x_s)^\top f(x_t)] \cdot [f(x_t)^\top f(x_s)] \right)^{1/2} \tag{18}$$

$$= \left( f(x_s)^\top \cdot \sum_{x_t \in X_T} [f(x_t)f(x_t)^\top] \cdot f(x_s) \right)^{1/2} \tag{19}$$

$$\propto \left[ f(x_s)^\top \left( \frac{1}{|X_T|} \sum_{x_t \in X_T} f(x_t)f(x_t)^\top \right) f(x_s) \right]^{1/2} \tag{20}$$

Note that $\Lambda = \frac{1}{|X_T|} \sum_{x_t \in X_T} f(x_t)f(x_t)^\top$ is the variance matrix of the target image embeddings. Then using NormSim$_2$ for filtering, we have

$$S = \arg\max_{|S|=N} \sum_{x_s \in X_S} \text{NormSim}_2(X_T, x_s) \tag{21}$$

$$\text{NormSim}_2(X_T, x_s) = f(x_s)^\top \cdot \Lambda \cdot f(x_s) \tag{22}$$

$$= f(x_s)^\top U \cdot S \cdot U^\top f(x_s) \tag{23}$$

$$= \sum_{j=1}^{r} s_j \cdot [f(x_s)^\top u_j]^2 \tag{24}$$

Here $\Lambda = USU^\top$ is the eigen decoposition of $\Lambda$, where $S = \text{diag}(s_1, \ldots, s_r)$ with $s_1 > \ldots > s_r$ are the matrix of eigenvalues, and $U = [u_1, \ldots, u_r] \in \mathbb{R}^{d \times r}$ are the corresponding eigenvectors (i.e., the principal component directions). Note that the column vectors of $U$ and $f(x_s)$ are all unit vectors, (24) shows that NormSim$_2$ select the data that match with the principal components, i.e., eigen directions $u_j$ with large eigen values $s_j$.

### C.6.2 Why NormSim works well without explictly considering data diversity.

We answer this question by the following reasons:

- Many top baselines, such as DFN and T-MARS, also don't explicitly consider diversity, yet they still provide good performance. Devil even shows that valuable data is worth sampling multiple times, which they call "quality duplication". Therefore, one important reason why NormSim works well without explicitly considering diversity may be that when the computing budget is limited, as in the DataComp benchmark, the model first needs to learn the most useful and representative data, which should be similar to some target data.

- Moreover, we chose validation data from 24 downstream tasks ranging from ImageNet to EuroSet, which may have covered a sufficiently diverse range of target examples for NormSim to calculate similarity. The diversity of the target data will consequently result in the diversity of the selected subset. And this also implies the importance of selecting a good target dataset.

- An additional reason may be that our proposed negCLIPLoss already implicitly selects more diverse data, as shown in Figure 1 of the main paper. If some training data are diverse, they will match less with other data and thus have a lower normalization term. This results in a larger negCLIPLoss and a higher probability of being sampled.

## D  Additional Results

### D.1  Stability Analysis of Batch Sampling Numbers in negCLIPLoss

We show that negCLIPLoss is not sensitive to the number of random select batches $K$ in Figure 5.

### D.2  Universality of negCLIPLoss over Different Teacher Models

We show the complete results of applying our methods to different teacher models like OAI CLIP-B/32 and DFN-P in Table 7. Detail descriptions are in Sec. 4.

### D.3  NormSim$_\infty$ is Better than Nearest Neighbor Selection

We also try to use near-neighbor selection for aligning downstream distribution. Here, we calculate the ranks of pretraining data for each target (the higher the rank, the higher the similarity), and then for each pre-train data, we keep its highest rank. Finally, we select the data with the highest ranks as the nearest neighbor selected subset.

In Table 8, we show that given the training data of 22 downstream tasks, our NormSim$_\infty$ can outperform near neighbor selection under the same downsampling ratio. The reason may be that the distribution between the target and pretraining set is not well aligned, so if you force the algorithm to

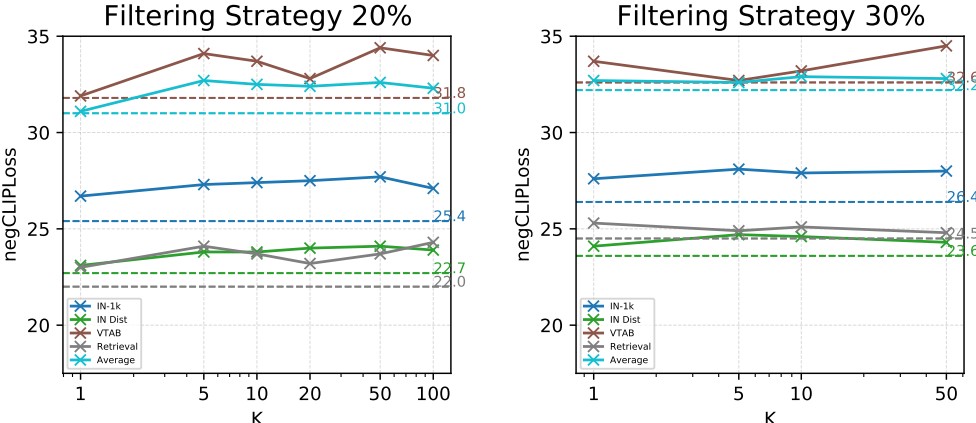

Figure 5: Results of negCLIPLoss with a different number of batch samples (denoted as $K$) on DataComp-medium. Solid lines denote negCLIPLoss, while dashed lines denote CLIPScore. Here, we use OAI CLIP-L/14 as the pretrained model. We can see that once $K \geq 5$, negCLIPLoss consistently outperforms CLIPScore across all subtask metrics. In the main paper, we set $K = 10$.

Table 7: Results on DataComp-medium from the top methods that use only OpenAI's CLIP-B/32 model or public version of DFN (DFN-P).

| OAI CLIP-B/32 | Dataset Size | IN-1k (1 sub-task) | IN Dist. Shift (5) | VTAB (11) | Retrieval (3) | Avg. (38) |
|---|---|---|---|---|---|---|
| CLIPScore (20%) | 22M | 27.0 | 23.8 | 33.0 | 22.9 | 32.2 |
| CLIPScore (30%) | 33M | 27.6 | 24.2 | 33.6 | 25.1 | 33.2 |
| negCLIPLoss (20%) | 22M | 28.9 | 24.8 | 34.3 | 24.3 | 33.0 |
| negCLIPLoss (30%) | 33M | 28.8 | 25.1 | 33.7 | 26.6 | 33.6 |
| negCLIPLoss (30%) $\cap$ NormSim$_\infty$(Target) | 22M | **32.4** | **27.4** | **35.9** | **26.3** | **35.2** |
| **DFN-P** | | | | | | |
| CLIPScore (15%) | 16M | 25.9 | 23.3 | 32.9 | 21.9 | 31.6 |
| CLIPScore (17.5%) | 19M | 30.2 | 26.8 | 34.1 | 26.5 | 33.8 |
| CLIPScore (20%) | 22M | 29.7 | 26.8 | 33.0 | 27.0 | 33.1 |
| CLIPScore (30%) | 33M | 28.4 | 24.7 | 33.2 | 26.8 | 32.7 |
| negCLIPLoss (15%) | 16M | 31.3 | 27.3 | 35.8 | 26.4 | 34.6 |
| negCLIPLoss (17.5%) | 19M | 31.2 | **27.5** | 35.7 | 27.0 | **34.7** |
| negCLIPLoss (20%) | 22M | 30.7 | 27.4 | 33.6 | **27.5** | 33.8 |
| negCLIPLoss (30%) | 33M | 28.9 | 25.5 | 33.4 | 27.3 | 33.2 |
| negCLIPLoss (30%) $\cap$ NormSim$_\infty$(Target) | 22M | 29.4 | 23.6 | 33.5 | 24.2 | 32.5 |
| negCLIPLoss (17.5%) $\cap$ NormSim$_\infty$(Target) | 16M | 31.5 | 26.4 | 34.6 | 25.4 | 34.4 |
| negCLIPLoss (17.5%) $\cap$ NormSim$_\infty^{B/32}$(Target) | 16M | **31.6** | 27.3 | **37.2** | 25.5 | **35.7** |

find the nearest train data for each target, that train data may be sometimes random and not helpful. On the other hand, NormSim$_\infty$ will not select this kind of data. It will select the data whose best similarity score exceeds some general threshold, rather than just consider ranks.

### D.4 Vision-Only NormSim is Better than Using Both Vision and Language

In DataComp [1], they show that image-based filtering is better than text-based filtering. In our paper, we also do an ablation study to support this. Due to the restriction of computation resources, we run our NormSim$_2$(IN-1k) and NormSim$_2$-D on DataComp-small as an example. Since ImageNet-1k

Table 8: Comparison between $\text{NormSim}_\infty$ and nearest neighbor selection. We use OAI CLIP-L/14 as the teacher model and assume both methods have been intersected with negCLIPLoss (30%). The size of the selected subset is 22M.

| Filtering Strategy | IN-1k | VTAB | Avg. |
|---|---|---|---|
| negCLIPLoss (30%) | 27.9 | 33.2 | 32.9 |
| Nearest Neibor Selection | 31.5 | 34.9 | 34.0 |
| $\text{NormSim}_\infty$(Target) | **31.7** | **36.0** | **35.0** |

only has labels rather than long texts for describing images, we need to generate the caption before calculating $\text{NormSim}_2$(IN-1k). We select 80 templates as the original CLIP paper [4], generate prompts for each class, and take the mean of their embeddings as the representative text embedding for images within that class.

The results are in Table 9. We can see that for both metrics, we have **"image only" > "image $\times$ text" > "text only"**. We believe the reason for $\text{NormSim}_2$(IN-1k) is that the images themselves can convey significantly more features than the text prompts generated by labels. For $\text{NormSim}_2$-D, it should be related to the large amounts of low-quality captions in the web-curated dataset. And "image $\times$ text" will also be influenced by the informativeness and the quality of captions. In short, for NormSim, using vision-only embeddings is a best choice.

Table 9: Ablation Study on the NormSim and its variants on DataComp-small (11M). All experiments first select 45% data based on the CLIP score, then use corresponding approaches to obtain 3.3M data. "image" or "text" means using the variance of image or text embeddings to represent $\bar{\Sigma}_{\text{target}}$, and "image $\times$ text" means representing $\bar{\Sigma}_{\text{target}}$ with the cross-covariance of image and text embeddings.

| Filtering Strategy $\cap$ CLIP score (45%) | IN-1k | IN Dist. Shift | VTAB | Retrieval | Average |
|---|---|---|---|---|---|
| Random Sampling | 4.2 | 4.9 | 17.2 | 11.6 | 15.6 |
| **NormSim** (IN-1k, image) | **5.2** | **5.5** | 19.0 | **12.2** | **17.4** |
| **NormSim** (IN-1k, text) | 3.9 | 4.2 | 16.3 | 11.3 | 14.9 |
| **NormSim** (IN-1k, image $\times$ text) | 4.3 | 4.9 | 17.5 | 11.8 | 15.9 |
| **NormSim-D** (image) | 4.7 | 5.4 | **19.7** | 11.7 | 17.3 |
| **NormSim-D** (text) | 3.5 | 4.1 | 16.7 | 11.1 | 15.4 |
| **NormSim-D** (image $\times$ text) | 3.6 | 4.2 | 18.4 | 11.1 | 15.8 |

# E    Additional Visualization

We further visualize[8] more data with different negCLIPLoss in Figure 6, 7 and 8. And similar for $\text{NormSim}_\infty$(Target) in Figure 9, 10 and 11.

---

[8]We use `https://github.com/ypwang61/research_tools/blob/main/visualization2.py` (ImageCaptionVisualizer) for visualizing the dataset. We also recommend visualizing basic dataset statistics by `https://lst627.github.io/visdatacomp.github.io/`.

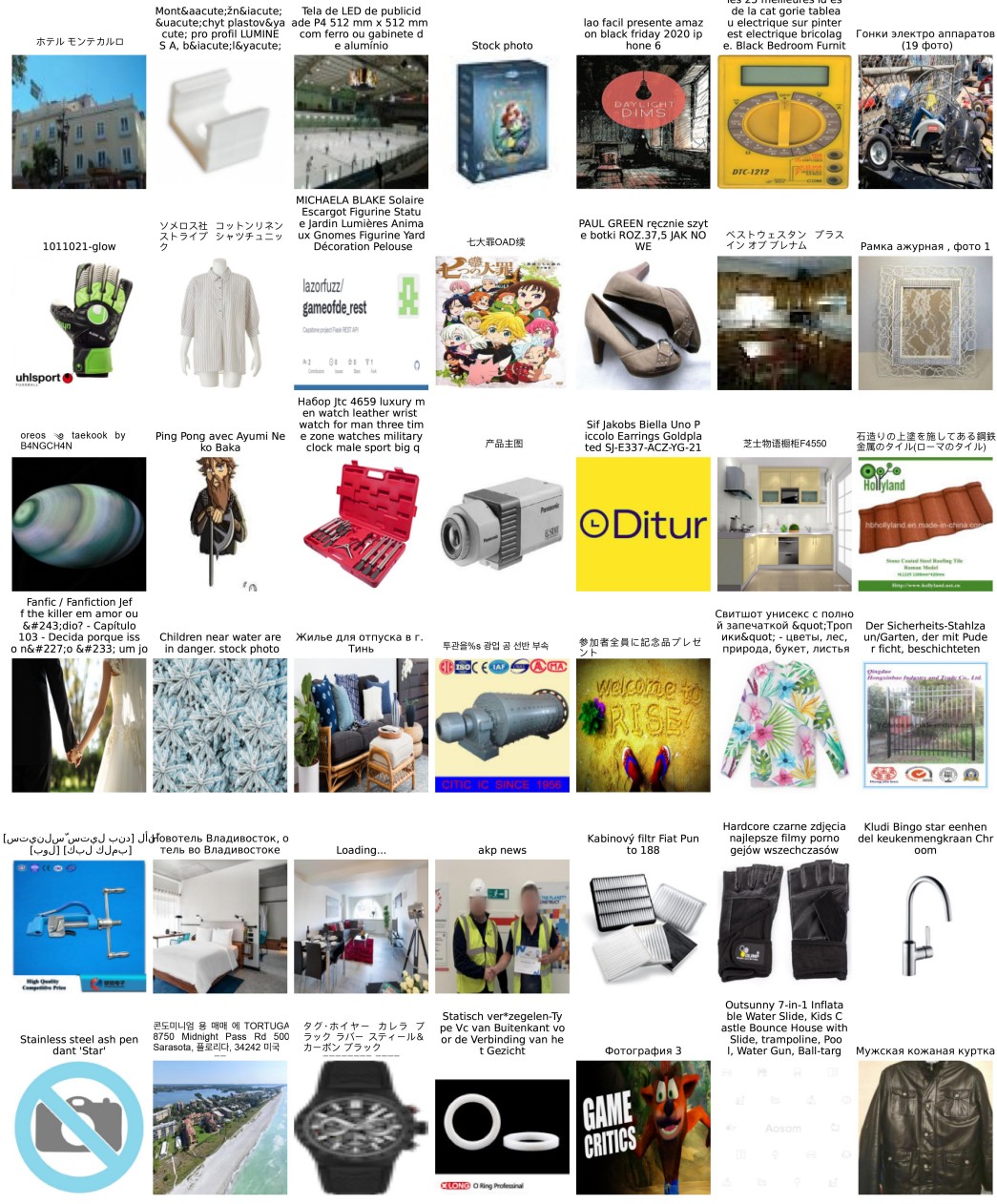

Figure 6: Visualization of a small subset whose negCLIPLoss rank top 100% high in DataComp-medium.

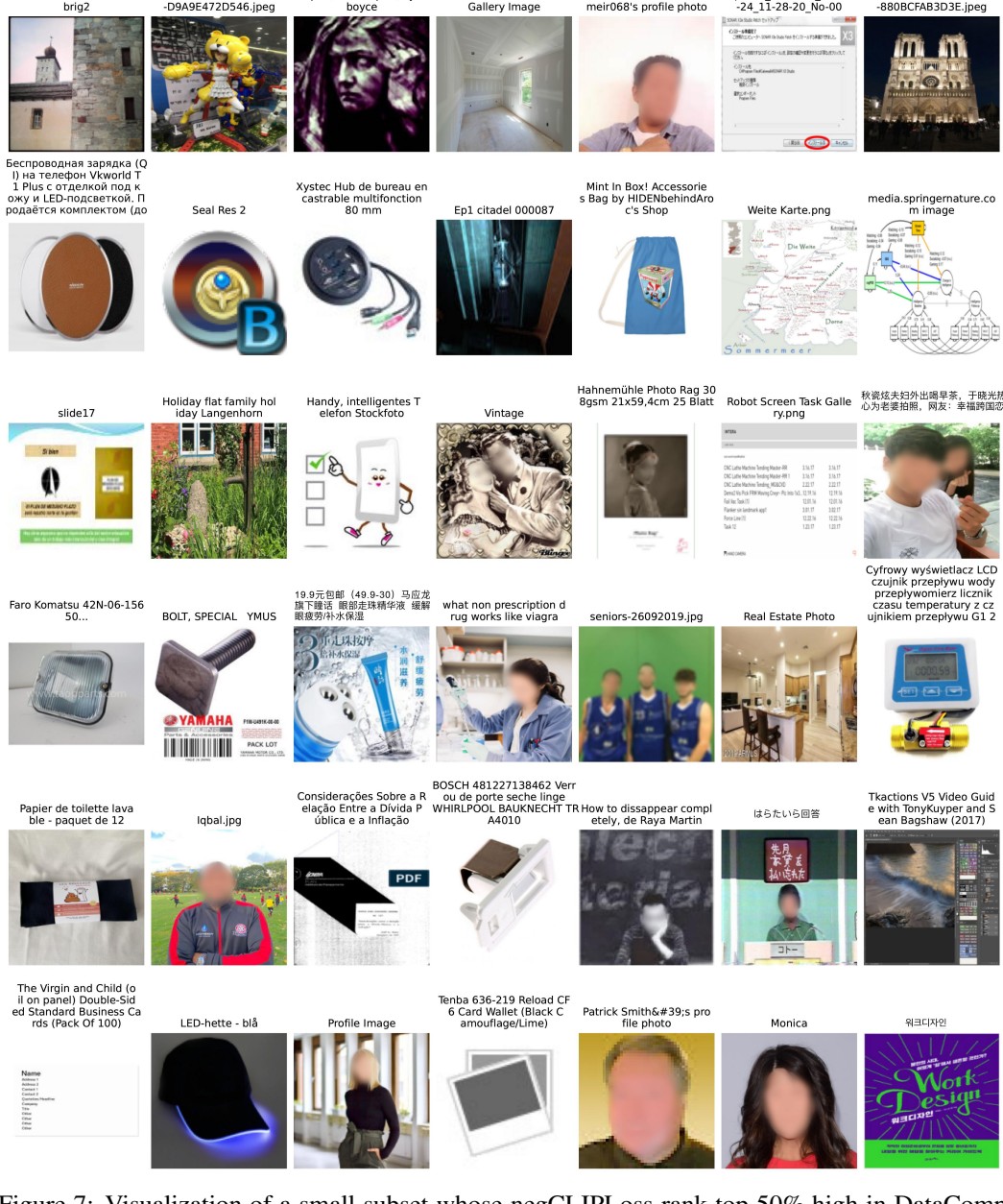

Figure 7: Visualization of a small subset whose negCLIPLoss rank top 50% high in DataComp-medium.

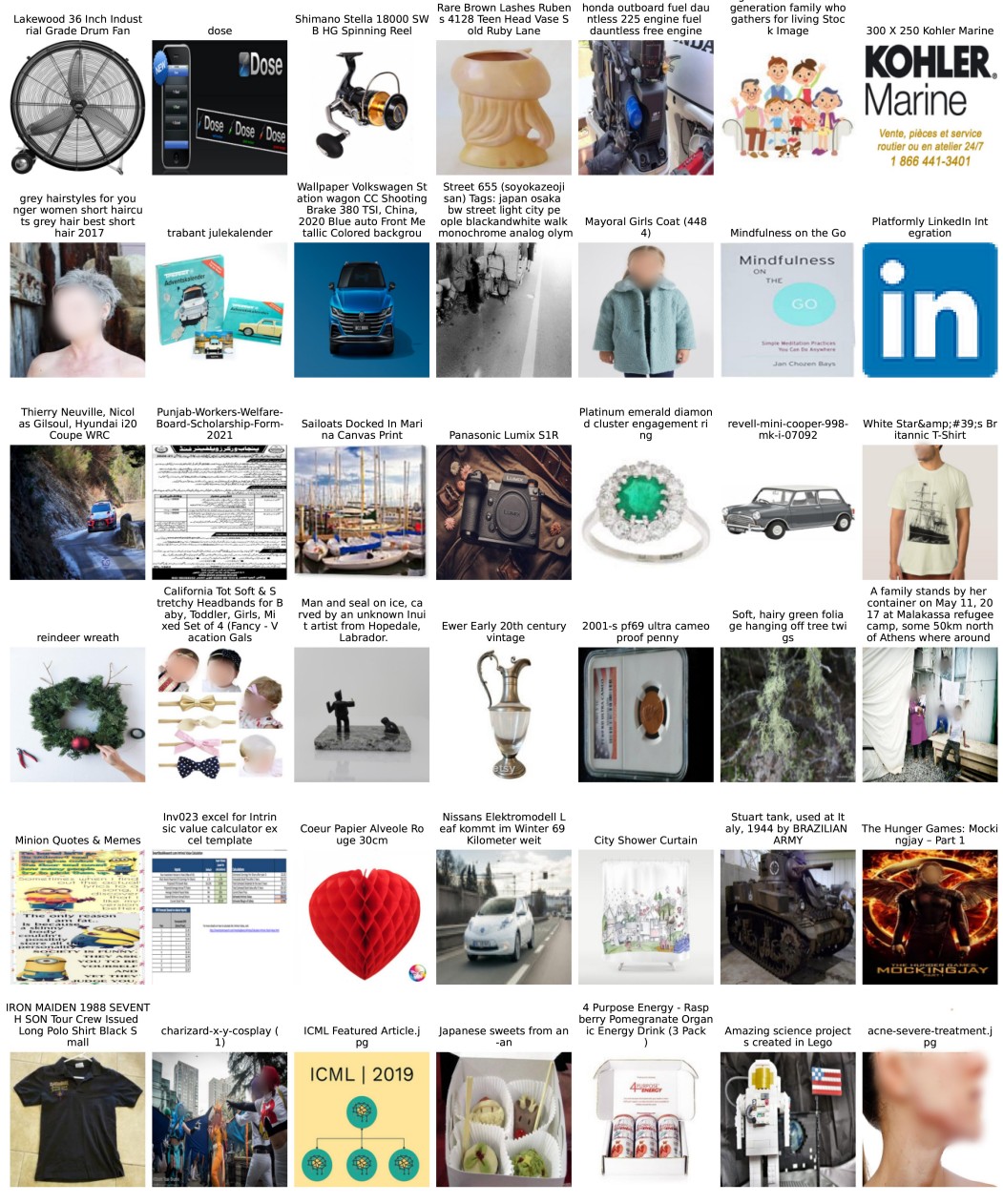

Figure 8: Visualization of a small subset whose negCLIPLoss rank top 10% high in DataComp-medium.

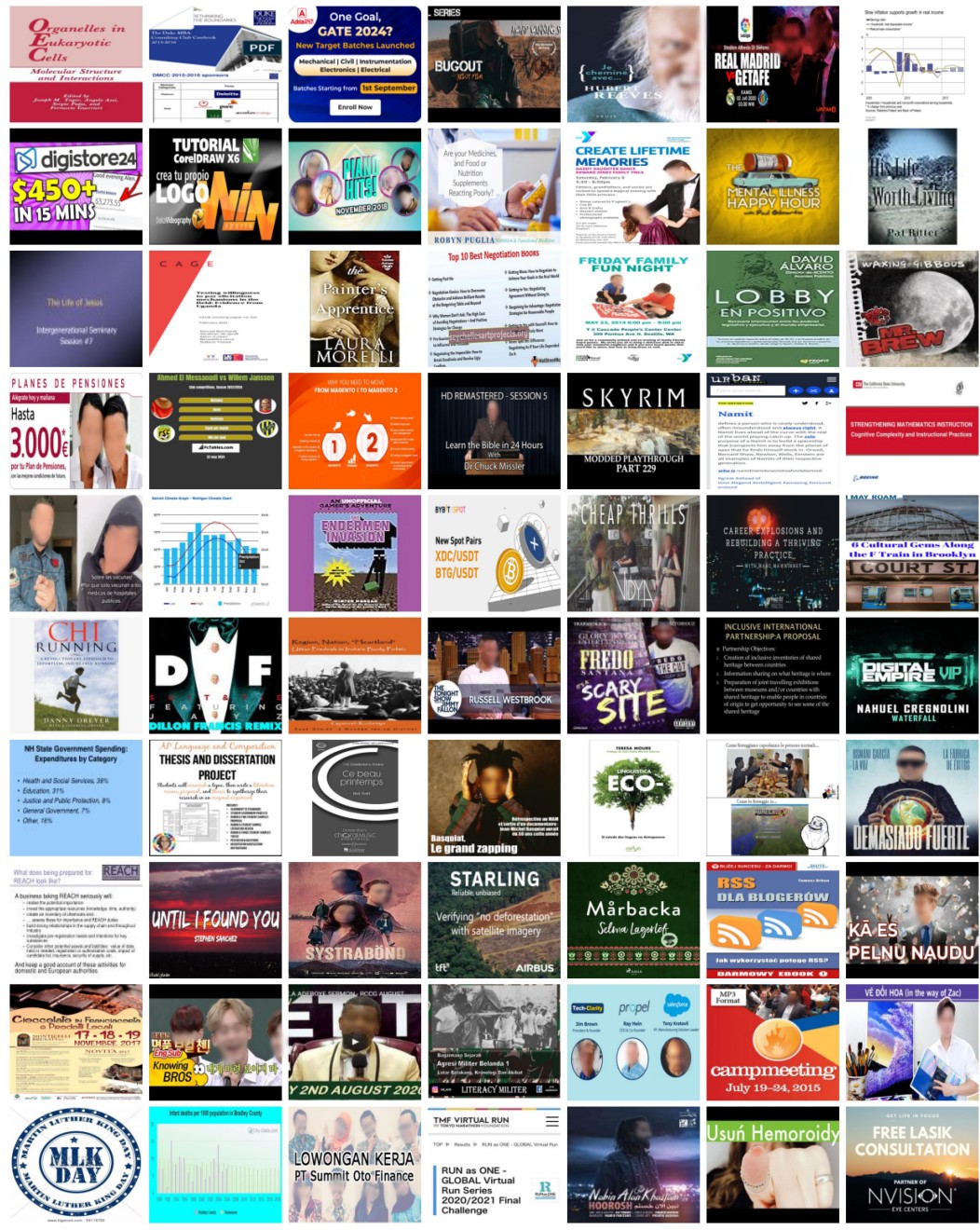

Figure 9: Visualization of the images from a small subset whose NormSim$_\infty$(Target) rank top 100% high in DataComp-medium.

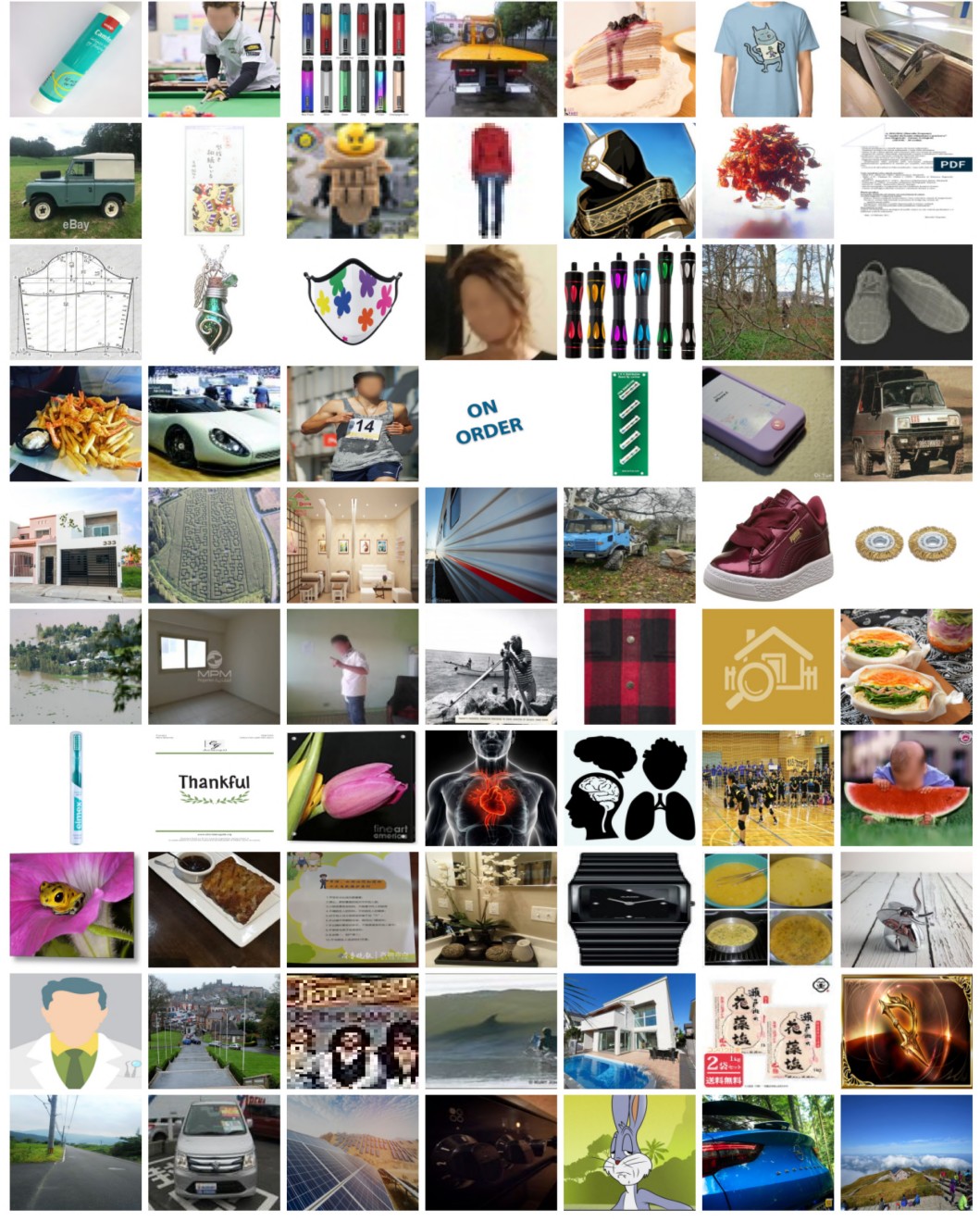

Figure 10: Visualization of the images from a small subset whose NormSim$_\infty$(Target) rank top 50% high in DataComp-medium.

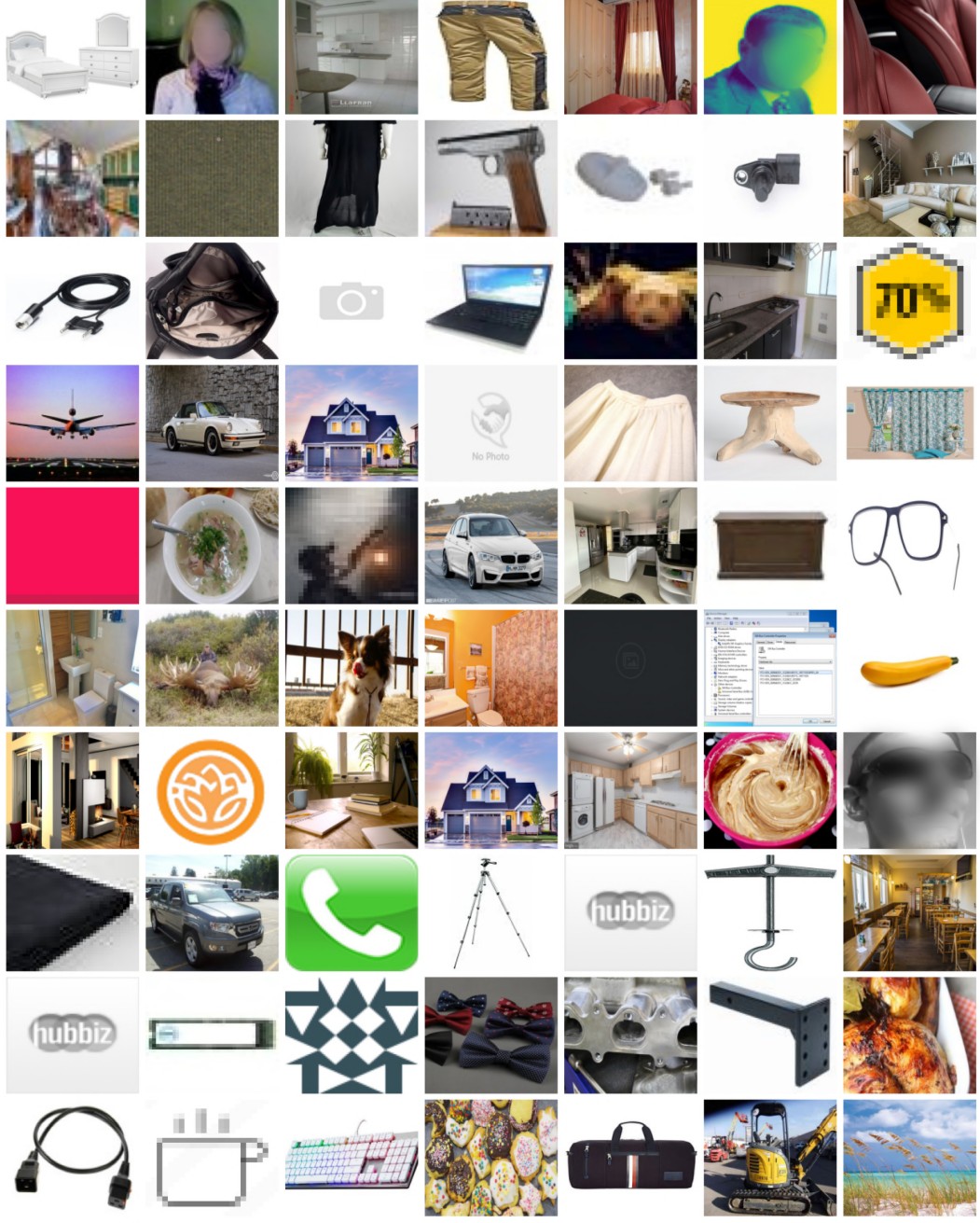

Figure 11: Visualization of the images from a small subset whose NormSim$_\infty$(Target) rank top 10% high in DataComp-medium.

