# OpenReview forum: "CLIPLoss and Norm-Based Data Selection Methods for Multimodal Contrastive Learning"
_NeurIPS.cc/2024/Conference — NeurIPS 2024 spotlight_

### Official Review · Reviewer_NSn3 · 2024-06-27

**Soundness:** 4
**Presentation:** 4
**Contribution:** 3
**Rating:** 7
**Confidence:** 4

**Summary:**

This paper introduces a novel metric, negCLIPLoss, for selecting high-quality data. Additionally, the paper proposes a norm-based metric, Normsim, which offers an improved measure of data quality and is compatible with existing methods. Both negCLIPLoss and NormSim demonstrate significant performance improvements, outperforming state-of-the-art methods, while maintain low preprocessing time. Theoretical interpretations are provided for NormSim within the framework of a linear model.

**Strengths:**

1. The paper is intuitive, well-motivated and well-written.
2. The proposed methods are simple and effective.
3. The experiments are sufficient.

**Weaknesses:**

1. In Line 86, the authors assert that NormSim does not explicitly consider diversity but provide no further explanation. Since diversity is often linked to the generalization performance of models, it is unclear how the proposed methods implicitly connect with diversity.

**Questions:**

1. The algorithm 1 requires the knowledge of the batch size B and the parameter $\tau$ from the teacher model. If the teacher model is private and both B and $\tau$ are not accessible (for example, only api is provided), are the proposed methods still workable? How critical is the batch size B to model performance? Can the parameter $\tau$ be estimated?

---

> ### Author Rebuttal · Authors · 2024-08-07
>
> Thank you for your recognition of our paper and your constructive feedback. We have responded to your concerns and will revise our paper based on the discussions. We would also appreciate it if you could let us know if our response addresses your concerns.
>
>
> > **Q1**: In Line 86, the authors assert that NormSim does not explicitly consider diversity but provide no further explanation. Since diversity is often linked to the generalization performance of models, it is unclear how the proposed methods implicitly connect with diversity.
>
>
> **A1**: We reply to this concern as the following points:
> 1) Many top baselines, such as DFN and T-MARS, also don't explicitly consider diversity, yet they still provide good performance. Devil [1] even shows that valuable data is worth sampling multiple times, which they call 'quality duplication'. Therefore, one important reason why NormSim works well without explicitly considering diversity may be that when the computing budget is limited, as in the DataComp benchmark, the model first needs to learn the most useful and representative data, which should be similar to some target data.
> 2) Moreover, we chose validation data from 24 downstream tasks ranging from ImageNet to EuroSet, which may have covered a sufficiently diverse range of target examples for NormSim to calculate similarity. The diversity of the target data will consequently result in the diversity of the selected subset.
> 3) An additional reason may be that our proposed negCLIPLoss already implicitly selects more diverse data, as shown in Figure 1 of the main paper. If some training data are diverse, they will match less with other data and thus have a lower normalization term $R$. This results in a larger negCLIPLoss and a higher probability of being sampled.
>
> Thanks for your concern and we would like to add these discussions into the NormSim section in the revised paper.
>
> [1] Yu, Haichao, et al. "The devil is in the details: A deep dive into the rabbit hole of data filtering." arXiv preprint arXiv:2309.15954 (2023).
>
> > **Q2**: Algorithm 1 requires the knowledge of the batch size B and the parameter $\tau$ from the teacher model. If the teacher model is private and both B and $\tau$ are not accessible (for example, only api is provided), are the proposed methods still workable? How critical is the batch size B to model performance? Can the parameter $\tau$ be estimated?
>
> **A2**: This is a good concern about the limitation of our method. First we claim that most of the CLIP model is either close-sourced (no API, like the SOTA filtering model of DFN) or fully open-sourced (providing model weights, like OAI CLIP, openclip, LAION, etc), so our method should be workable for most of the current CLIP models. Besides, when only the API is provided, the recommended values for $B$ and $\tau$ are 32768 and 0.01, respectively. The reason is that
>
> 1) In general, similar to the training stage, a larger batch size can result in better performance in negCLIPLoss filtering since it contains more contrastive data pairs in a batch. 32768 is the training batch size of the OAI CLIP model, and these data can be fitted into a single 24G GPU in the CLIP forward pass. In A1 in the ‘reply to all reviewers’ parts, we also theoretically show that using a larger batch size, negCLIPLoss will have a smaller approximation error.
>
> 2) For $\tau$, when the model is accessible, we can directly get it by the model parameters since it’s learnable (And note that their temperature is the reciprocal of our definition). However, their values of $\tau$ are always as regular as 0.01. The reason is that there is a manually set lower bound in the CLIP training setup (for the original definition it’s upper bound 100) for the trainable $\tau$, and after training they always reach this bound. Therefore, when the model parameter is unavailable, we recommend first trying 0.01 for $\tau$, and then sampling a small subset and tuning $\tau$ around it. For tuning the parameters, except for training a small-scale model, we also recommend sampling a small subset and calculating the negCLIPLoss on it with different hype-parameters settings, and then visualizing them (like Figure 6-11 in the main paper) for choosing. Details are shown in Appendix C.5.
>
> What’s more, to show how batch size B influences the model performance, we do some ablation study on $B$ and $\tau$. Due to the limited time and resources, we mainly focus on the OAI CLIP-B/32 model. The results are shown in A2 in the ‘reply to all reviewers’ parts (Table R1). In R1 we can see that in general, |B| = 32768 is better than |B|=16384, and $\tau=0.01$ performs the best for both batch sizes. These results support our claims above. We will add these ablation studies and the discussion in the revised paper.

---

### Official Review · Reviewer_XD1b · 2024-07-12

**Soundness:** 4
**Presentation:** 4
**Contribution:** 3
**Rating:** 8
**Confidence:** 3

**Summary:**

This work proposes two new approaches for data selection for vision-language pre-training. The first approach, negCLIPLoss, adds the contrastive loss as a normalization term on top of the existing CLIPScore metric. The second approach, NormSim, further improves the performance if examples from target task distribution are available. Together, the methods achieve state-of-the-art results on ImageNet-1K and 38 downstream tasks with DataComp-medium without any external data or model. Several important theoretical justifications and interpretations are provided for the methods.

**Strengths:**

Quality: the quality is overall high. Without external resources (on which previous methods rely), the proposed approaches improve evaluation performances by 5.3% on ImageNet and 2.8% on average of 38 downstream tasks. Further, there are many theoretical results that focus on the guarantees of NormSim (though with strong assumptions).

Clarity: this paper is very well written. It is well-motivated, the distinctions of previous approaches are succinctly laid out, the methods are well presented, and the results have a clear structure.

Significance: this paper will bring significant impacts. The data selection problem has been increasingly vital for training higher-quality vision-language models. The proposed approaches, which focus on metrics instead of models or data, are compatible with different techniques that can be combined with advanced models in the future. The approaches also provide significant efficiency improvements (e.g., from 70 L40 hours to 5 L40 hours). The theoretical analyses can provide useful tools for future research as well.

**Weaknesses:**

Quality: this is a minor complaint, but in Lines 229 - 230 the authors state that "the results of baselines on the leaderboard do not apply to our datasets, and we reproduce all the top baselines on the leaderboard with their public UIDs of the selected data" because some URLs of images become invalid. The leaderboard scores of baselines seem higher than the reproduced results in the submission. Could the authors also include the DataComp leaderboard results in the Appendix for fair comparison?

There are also some minor questions below.

**Questions:**

1. In Lines 135 - 136, the inaccessible batch division $B^*$ from teacher CLIP models is different from the actual batch $B_k$ in this work, in terms of both the actual image-text pairs and the batch size. Are there any potential theoretical guarantees or approximations to show that such a difference is reasonably negligible?

2. Could the authors further show the derivations of the discussions on the two important NormSim instances? 1) Lines 179 - 180 ($p=2$, equivalent to selecting a subset that aligns with the principal components), and 2) Lines 181-182 ($p=\infty$, a sample will be selected if it has high similarity to any target)? These may help other readers to understand NormSim better.

**Limitations:**

The authors discussed the limitations of this work.

---

> ### Author Rebuttal · Authors · 2024-08-07
>
> Thank you for your recognition of our paper together with your valuable comments and suggestions. We will revise our paper according to your comments. We respond to your questions below and would appreciate it if you could let us know if our response addresses your concerns.
>
>
> > **Q1**: this is a minor complaint, but in Lines 229 - 230 the authors state that "the results of baselines on the leaderboard do not apply to our datasets, and we reproduce all the top baselines on the leaderboard with their public UIDs of the selected data" because some URLs of images become invalid. The leaderboard scores of baselines seem higher than the reproduced results in the submission. Could the authors also include the DataComp leaderboard results in the Appendix for fair comparison?
>
> **A1**: Thanks for your advice! We would include the DataComp leaderboard results in appendix in the revised version.
>
> > **Q2**: In Lines 135 - 136, the inaccessible batch division B∗ from teacher CLIP models is different from the actual batch Bk in this work, in terms of both the actual image-text pairs and the batch size. Are there any potential theoretical guarantees or approximations to show that such a difference is reasonably negligible?
>
> **A2**: Thanks for mentioning this. We construct a theorem using the concentration inequality to show that when the batch size is sufficiently large, the normalization term $R^{B_k}$ obtained from actual batch $B_k$ can approximate $R^{B^*}$ calculated using ground truth batch $B^*$ quite well, i.e., $R^{B_k} = (1+o(1))R^{B^*}$. The details have been shown in A1 in the ‘reply to all reviewers for the major concern’ parts. Here we assume that $B^*$ and $B_k$ are i.i.d. for simplicity since the claim cannot hold if the teacher batch is very different from the actual batch. We also assume that $|B|=|B^*|$. In practice, we claim that a larger batch size is better since it can contain more contrastive pairs in a batch, and we do some ablation studies as shown in A2 in the ‘reply to all reviewers for the major concern’ parts (Table R1) to support our claim.
>
>
> > **Q3**: Could the authors further show the derivations of the discussions on the two important NormSim instances? 1) Lines 179 - 180 (p=2, equivalent to selecting a subset that aligns with the principal components), and 2) Lines 181-182 (p=∞, a sample will be selected if it has high similarity to any target)? These may help other readers to understand NormSim better.
>
> **A3**: Thanks for your advice, we show the derivations as follows and we add them in the revised paper. For convenience, we let $f(x_t)$ denote the image embedding of the target data $x_t \in X_T$, and $f(x_s)$ denotes the image embeddings of training data $x_s \in X_S$. Then the definition of NormSim on a data $x_s$ is
>
> $$
> NormSim_p(X_{T}, x_s) =  \left(\sum_{x_t \in X_T}  [f(x_t)^\top f(x_s)]^p\right)^{1/p} \qquad (R1)
> $$
> Then when $p=2$, we have
> $$
> NormSim_2(X_{T}, x_s) = \left(\sum_{x_t \in X_T} [f(x_s)^\top f(x_t)]\cdot [f(x_t)^\top f(x_s)] \right)^{1/2} = \left(f(x_s)^\top \cdot\sum_{x_t \in X_T} [f(x_t) f(x_t)^\top ]\cdot f(x_s) \right)^{1/2}
> $$
> Note that $\Lambda=\frac{1}{|X_T|}\sum_{x_t \in X_T} [f(x_t) f(x_t)^\top]$ is the variance matrix of the target image embeddings. Then using $NormSim_2$ for filtering, we have
>
> $$
> S = \arg \max_{|S|=N}\sum_{x_s \in X_S} NormSim_2(X_{T}, x_s) = \arg \max_{|S|=N}\sum_{x_s \in X_S}  f(x_s)^\top \cdot \Lambda \cdot f(x_s) \qquad (R2)
> $$
> Take $\Lambda=USU^\top$ as the eigen decoposition of $\Lambda$, $S = \text{diag}(s_1,\ldots,s_r)$ where $s_1>\ldots > s_r$ is the matrix of eigenvalues, and $U=[u_1,\ldots,u_r] \in R^{d\times r}$ are the corresponding eigenvectors, i.e., the principal component directions.
> Note that the column vectors of $U$ and $f(x_s)$ are all unit vectors, so we get that Eqn. R2 means $\text{NormSim}_2$ select the data that best match with the principal components of the target variance.
>
> Besides, when $p=\infty$, from Eqn. R1 and the definition of infinity norm, we know that $NormSim_{\infty}(X_{T}, x_s) = \max_{x_t \in X_T} f(x_t)^\top f(x_s)$, thus it measures the max similarity between the data $x_s $ with any target data $x_t \in X_T$. Therefore, a sample will be selected if it has high similarity to any target data.
>
> We will add these discussions in the revised papers.

---

> ### Comment · Reviewer_XD1b · 2024-08-09
> **Response**
>
> The reviewer thanks the authors for the global and the specific responses. The reviewer is satisfied with the response and will maintain the score.

---

> > ### Author Response · Authors · 2024-08-09
> > **Thank you for reviewing**
> >
> > We sincerely thank you for your time and constructive advice on improving our work!

---

### Official Review · Reviewer_3xCf · 2024-07-14

**Soundness:** 2
**Presentation:** 2
**Contribution:** 3
**Rating:** 6
**Confidence:** 5

**Summary:**

Data selection is crucial in the pretraining stage to clean the web-crawled, large, and noisy pretraining dataset. Typically, existing methods use embeddings to compute CLIPscore in order to assess the data sample alignment quality. This paper introduces two new methods to enhance this measurement:
1. negCLIPLoss: a better adjustment to reduce bias within a given batch.
2. NormSim: provides additional information when downstream tasks are known, allowing the selection of samples that are close to the target downstream tasks.
Empirical results demonstrate that these proposed methods can be easily combined with existing filtering approaches. The authors also illustrate that their approach yields state-of-the-art results on the DataComp leaderboard.

**Strengths:**

* Originality: Most of the work in data curation relies heavily on the original CLIP score. It's a new idea to adapt the CLIP score and elevate this measurement for better use.
* Quality: The resulting performance is solid and achieves the top position on the leaderboard (medium-scale).
* Clarity: The motivation behind the two approaches is clear, but some areas need further clarification. Questions are listed below.
* Significance: Data selection in the pretraining dataset is important to the field, and they have demonstrated that their approaches are effective in achieving state-of-the-art results.

**Weaknesses:**

1. I think we need more clarification on how to interpret the Top X% in three different metrics in Figure 1. Can the authors provide a more detailed description? Also, how is the R score derived from the batched data? How to find the proper batched data to use?
2. It seems that the negCLIPLoss is not incorporated into the training loss. We use it as a measurement when CLIP embeddings are provided. In this scenario, how do we determine the batch data, B, for subtracting the regularization term? Would the size of the batched data affect the measurement? The sampling method to find batched data is unclear to me.
3. I am unclear about the process for greedily selecting samples using NormSim, especially when the raw data pool is massive, and how to define the size of S.
4. I would suggest moving algorithm steps from the Appendix into the main body, or showing some steps in the main body. They are good at understanding filtering steps.

**Questions:**

1. In Figure 1, R scores on the left side are in the top 100%, while on the right side they are in the top 10%. How should these be interpreted and categorized as underestimates or overestimates of quality?
2. When the downstream targets are not accessible, we may use the current filtered dataset as a reference, but how do we find the first-round reference dataset as a proxy to compute NormSim?
3. I would like to list several papers that I found and read for data selection.

https://arxiv.org/abs/2405.15613, https://arxiv.org/abs/2401.12225, https://arxiv.org/abs/2302.03169, https://arxiv.org/abs/2401.04578, https://arxiv.org/abs/2404.07177

**Limitations:**

I didn't see any potential negative societal impact of their work.

---

> ### Author Rebuttal · Authors · 2024-08-07
>
> Thank you for your constructive feedback to help us improve our paper. We will revise our paper based on your feedback. We detail our response below and please kindly let us know if our response addresses your concerns.
>
> > **Q1**: I think we need more clarification on how to interpret the Top X% in three different metrics in Figure 1. Can the authors provide a more detailed description? (How should these be interpreted and categorized as underestimates or overestimates of quality?)
>
> **A1**: Thanks for mentioning this. We show the modified Figure 1 in the one-page supplementary based on your advice, and we illustrate it in detail as follows.
> We use the ‘Top X%’ of a metric to denote the score which is top X% high among all the scores of this metric in the data pool. For example, in Figure 1, R scores on the left side are top 100%, indicating that these examples have the smallest R in the dataset.
>
> Besides, in this case, we note them as ‘CLIPScore can underestimate the quality’, mainly just because their CLIPScore is relatively small (like Top 78%) while negCLIPLoss are high (like Top 34%). As we can see from both visualization and the experimental result, those data have high quality which is underestimated by CLIPScore. Similar claims hold overestimation cases. In Lines 154-165 in the main paper, we further show the intuition behind the normalization term $R$.
>
> > **Q2**:  How is the R score derived from the batched data? How to find the proper batched data to use?
>
> **A2**: We summarize how we choose random batch and obtain R score and negCLIPLoss from different batched data as follows:
>
> (1) We split the whole data into batches randomly, from which we obtain batches $\{B_1,\ldots, B_k\}$.
>
> (2) For each batch $B_s$, we calculate the cross-image-text similarity between the data in the batch, i.e., $f_l(x^l_i)^\top f_v(x^v_j)$ for any $i, j \in B_s$.
>
> (3) Using these scores, we can calculate the metrics of all the data in this batch from Eqn.1-2, and we record them for each data.
>
> (4) Repeat (1) - (3) for K times (Note each data will have multiple $R$ calculated from K different batches which all contain the data itself), we then calculate the mean value for these K different R scores and negCLIPLoss, and use them to approximate the ground-truth values.
>
> Details can be found in Algorithm 1 in Appendix C.1. We mention that this process isn’t the only choice to get the random batch. We choose this method mainly to avoid double calculation of the cross-image-text similarities.
>
> > **Q3**:  negCLIPLoss is not incorporated into the training loss. We use it as a measurement when CLIP embeddings are provided...Would the size of the batched data affect the measurement?
>
> **A3**: Yes, we use negCLIPLoss only for data filtering rather than training. We want to emphasize that the main focus of our paper is on data selection with fixed training pipelines. In A2 in the 'reply to all reviewers' parts, we show how the batch size affects the measurement. In A1 in the ‘reply to all reviewers’ parts, we also theoretically show that using a larger batch size, negCLIPLoss will have a smaller approximation error.
>
> > **Q4**: the process for greedily selecting samples using NormSim, especially when the raw data pool is massive
>
> **A4**: We note that NormSim is only determined by each data itself like CLIPScore, so the ‘greedily selecting samples using NormSim’ just means simply selecting the data with top NormSim scores. We use the words ‘greedily’ because for this particular NormSim-D algorithm (Details in Algorithm 2 in Appendix C.3), theoretically, we should solve harder optimization problems, but here we use a greedy way (select the top scores) to do approximation. In the revised paper we would change the word to prevent confusion.
>
> > **Q5**: how to define the size of S
>
> **A5**: In general, for all the top filtering methods, like CLIPScore, HYPE, and T-MARS, we always need to set the target size of the filtered dataset manually. Like in DataComp, all these top baselines keep the downsampling ratios ranging from 15%~30%. Our method with OAI CLIP first selects the data with the top 30% negCLIPLoss and then selects the top 66.7% NormSim scores to keep 20% of the original pool. We don’t tune the target size carefully here for fair comparison.
>
> In practice, this remains an open problem for all leading baselines when dealing with a large raw data pool. Here we found that a simple but very useful way to define $S$, is random sampling a small subset (like 1000 data) from the large pool and visualizing these data based on their scores, as Figure 6-11 in the main paper. From this we can determine the filtering threshold of the metric scores and thus the target size. (like we find 0.7~0.75 is a good threshold for NormSim). Details are shown in Appendix C.5.
>
> But overall, deciding a proper S is beyond the scope of this paper. We agree that this can be a meaningful direction for future research. We are also aware of some recent works [1] that suggest there are scaling laws for data filtering, indicating that the target size for filtering is strongly dependent on the computing budget.
>
> [1] Goyal, Sachin, et al. "Scaling Laws for Data Filtering--Data Curation cannot be Compute Agnostic." Proceedings of the IEEE/CVF Conference on Computer Vision and Pattern Recognition. 2024.
>
> > **Q6**: I would suggest moving algorithm steps from the Appendix into the main body
>
> **A6**: Thanks for your advice! We will add them in the revised version.
>
> > **Q7**: how do we find the first-round reference dataset as a proxy to compute NormSim in NormSim-D?
>
> **A7**: For the first run, we just use the whole original dataset as the proxy for calculating $\text{NormSim}_2$. For effectiveness, we only randomly downsample 10% of data for calculating $\text{NormSim}_2$, and the results are similar to using all the data.
>
> > **Q8**: list several related papers
>
> **A8**: Thanks for your advice! We would cite all these papers in the revised version.

---

> > ### Comment · Reviewer_3xCf · 2024-08-13
> >
> > Dear Authors,
> >
> > I have read your general response and individual comments. Thanks for your reply. Thanks for addressing studies on batch size and clarifying some details in the paper. In general, this paper gives a new idea and a good adjustment to replace CLIPScore, but some places lack detailed descriptions. I support this paper and keep my original score here. Thanks.

---

> > > ### Author Response · Authors · 2024-08-13
> > >
> > > Thanks for your reply and support, and we will add the suggested details mentioned in the rebuttal in the next version. Thanks for taking the time to make our paper better!

---

### Author Rebuttal · Authors · 2024-08-07

# Reply to all reviewers for the major concern

We sincerely appreciate all reviewers for their insightful and constructive feedback to make our paper better. We will revise our paper according to these comments. Here we will address the most common concerns of the reviewers and will put other responses in separate rebuttals.

Most of the reviewers have some concerns related to whether there is any (theoretical) guarantee that we can use the random batch from the pretraining dataset to approximate the inaccessible ground-truth batch in calculating $\mathcal{R}$ and negCLIPLoss, and how the batch size and temperature will affect our method negCLIPLoss. We answer these questions as follows

> **A1**: Concentration of Normalization Term $\mathcal{R}$

We construct a theorem using the concentration inequality to show that when the batch size is sufficiently large, the normalization term $R^{B_k}$ obtained from actual batch $B_k$ can approximate $R^{B^*}$ calculated using ground truth batch $B^*$ quite well. The details are as follows:

We assume that the pretraining dataset $\mathcal{D}$ is *i.i.d.* sampled from distribution $\mathcal{P}$. Besides, to use pretraining data batch to approximate the ground truth batch, one necessary condition is that their distribution is similar. Here for simplicity, we assume that they are also *i.i.d.*.

**Assumption R1**: We assume that the ground-truth batch of data $B^*$ used by the teacher model is *i.i.d.* to the pretraining dataset $\mathcal{D}$ which is required to be filtered.

For simplicity, we denote $s_{ij} = \bar f_{v}(x^v_i)^\top \bar f_{l}(x^l_j), i, j \in B$ to be the cross-image-text similarities in the batch $B$. Then the normalization term can be written as $\mathcal{R}^B_i = \frac{\tau}{2}\left[\log(\sum_{j \in B} \exp(s_{ij}/\tau)) + \log(\sum_{j\in B}\exp(s_{ji}/\tau))\right]$.
Here note that $s_{ij} \in [-1,1]$. We show that $\mathcal{R}_i^B = (1+o(1))\mathcal{R}_i^{B^*}$ for all $i$ when $|B|$ is sufficiently large, which means that we can use the random batch to approximate the ground-truth batch.

**Theorem R1**: If Assumption R1 holds and the batch size satisfies $|B|=|B^*|$, then we have $\mathcal{R}_{i}^B=\Theta(\log(|B|))$ while $|\mathcal{R}_i^B - \mathcal{R}_i^{B^*}| = O(\frac{1}{\sqrt{|B|}})$ for any $i \in B \cap B^*$.

*Proof*:
Since $s_{ij} \in [-1,1]$, It's obvious that $\mathcal{R}_i^B=\Theta(\log(|B|))$.

Let $\alpha_{ij} := e^{(s_{ij}/\tau)} - E_j[e^{(s_{ij}/\tau)}]$, then $\alpha_{ij}$ is zero-mean.
Note that since the data is *i.i.d.*, so does $\alpha_{ij}$.
Therefore, we denote $\gamma := E_{j}[\alpha_{ij}^2]$.
Note that $|\alpha_{ij}|\leq e^{1/\tau} =: M$, from Bernstein inequality we have
$$
    \mathbb{P}(|\sum_{j \in B}\alpha_{ij}| \geq t) \leq 2\exp(-\frac{\frac{1}{2}t^2}{|B|\gamma + \frac{1}{3}Mt})
$$
A similar conclusion holds for $B^*$. These result that with probability at least $1-\eta$, we have
$$
|\sum_{j \in B}\alpha_{ij}| \leq \max \left( 2\sqrt{|B|\gamma\ln(\frac{2}{\eta})}, \frac{4}{3}M\ln(\frac{2}{\eta}) \right) =: t(|B|,\gamma, \eta, M)
$$
Thus we have $|\sum_{j\in B}\exp(\frac{s_{ij}}{\tau})-\sum_{j\in B^*}\exp(\frac{s_{ij}}{\tau})| \leq 2 t(|B|,\gamma, \eta)$. Furthermore, for any $x_1, x_2 > 1$, it's easy to prove that $|\log(x_1)-\log(x_2)| \leq \frac{|x_1 - x_2|}{\min(x_1, x_2)}$. Therefore, we have $|\log(\sum_{j\in B}\exp(\frac{s_{ij}}{\tau}))-\log(\sum_{j\in B^*}\exp(\frac{s_{ij}}{\tau}))| \lesssim O(\frac{1}{\sqrt{|B|}})$, and thus similar claims hold for $|\mathcal{R}_i^B - \mathcal{R}_i^{B^*}|$.

> **A2**: Ablation study on batch size and the temperature.

All the reviewers are concerned about the choice of batch size. We claim that in general, similar to the training stage, **a larger batch size always results in better performance in negCLIPLoss filtering** since it can contain more contrastive data pairs in a batch, and thus it can check the image-text matching between more different data. Therefore, we consider the largest batch size 32768 which can fit into a single 24G GPU in the CLIP forward pass, and we note that this is also the training batch size that OpenAI used for training CLIP.

To support our claim, we do some ablation studies on $B$ and $\tau$. Due to the limited time and resources, we mainly focus on the OAI CLIP-B/32 model. Results are as in Table R1:

**Table R1**: Ablation study of $B$ and $\tau$ using OpenAI CLIP-B/32 model on DataComp-medium.

|  negCLIPLoss           | Dataset Size | ImageNet (1)   | ImageNet Dist. Shift (6) | VTAB (11) | Retrieval (3) | Avg. (38) |
|---------------|---------------|-------|------------|----|----|-----|
| $\|B\|=16384, \tau=0.01$| 33M | **28.8**| 25.0 | 32.5 | 26.2 | 33.0 |
| $\|B\|=16384, \tau=0.02$| 33M | 28.6 | 24.8 | 33.3 | 25.3 | 33.1 |
| $\|B\|=16384, \tau=0.07$| 33M | 28.0 | 24.2 | 33.5 | 25.1 | 32.6 |
| $\|B\|=32768, \tau=0.005$| 33M | 28.5 | 25.0 | 33.6 | **27.0** | 33.0|
| $\|B\|=32768, \tau=0.01$| 33M | **28.8** | **25.1** | **33.7** | 26.6 | **33.6**|
| $\|B\|=32768, \tau=0.02$| 33M | 28.5 | 24.8 | 33.6 | 26.2 | 32.9|
| $\|B\|=32768, \tau=0.07$| 33M | 28.2 | 24.5 | 32.8 | 25.2 | 32.7|
|  **negCLIPLoss $\cap$ NormSim**           |  |    |  |  |  | |
| $\|B\|=16384, \tau=0.01$| 22M | **32.4** | **27.4** | 34.5 | 26.1 | 34.7|
| $\|B\|=16384, \tau=0.02$| 22M | 31.8 | 26.7 | 35.0 | 24.9 | 34.2|
| $\|B\|=16384, \tau=0.07$| 22M | 31.0 | 26.3 | 35.0 | 25.5 | 33.9|
| $\|B\|=32768, \tau=0.005$| 22M | 32.2 | 27.2 | 35.3 | **26.5** | 34.8|
| $\|B\|=32768, \tau=0.01$| 22M | **32.4** | **27.4** | **35.9** | 26.3 | **35.2**|

We can see that in general, negCLIPLoss with a larger batch size ($|B|=32768$) indeed has better or comparable downstream performance. Nevertheless, $|B|=16384, \tau=0.01$ still has good performance when being combined with NormSim ($\tau=0.01$ performs well for both batch sizes). These match our theoretical findings in A1: using a larger batch size, negCLIPLoss will have a smaller approximation error.

---

### Decision · Program_Chairs · 2024-09-25

**Decision:**

Accept (spotlight)

**Comment:**

The submission proposes two data selection methods for large-scale visual-language model pretraining (e.g., CLIP). All three reviewers agree that this submission is well-written and intuitive with extensive experiments. The AC therefore recommends accepting the paper and asks the authors to include their discussions with the reviewers in the final manuscript.